# From regional to local SPTHA: efficient computation of probabilistic tsunami inundation maps addressing near-field sources

Manuela Volpe, Stefano Lorito, Jacopo Selva, Roberto Tonini, Fabrizio Romano, and Beatriz Brizuela

Istituto Nazionale di Geofisica e Vulcanologia, Italy

**Correspondence:** M. Volpe (manuela.volpe@ingv.it)

**Abstract.** Site-specific Seismic Probabilistic Tsunami Hazard Analysis (SPTHA) is a computationally demanding task, as it requires in principle a huge number of high-resolution numerical simulations for producing probabilistic inundation maps. We implemented an efficient and robust methodology using a filtering procedure to reduce the number of numerical simulations needed, while still allowing full treatment of aleatory and epistemic uncertainty. Moreover, to avoid biases in the tsunami hazard assessment, we developed a strategy to identify and separately treat tsunamis generated by near-field earthquakes. Indeed, the coseismic deformation produced by local earthquakes necessarily affects the tsunami intensity, depending on the scenario size, mechanism, and position, as coastal uplift or subsidence tend to diminish or increase the tsunami hazard, respectively. Therefore, we proposed two parallel filtering schemes in the far- and the near-field, based on the similarity of offshore tsunamis and hazard curves and on the similarity of the coseismic fields, respectively. This becomes mandatory as offshore tsunami amplitudes can not represent a proxy for the coastal inundation in case of near-field sources. We applied the method to an illustrative use-case at the Milazzo oil refinery (Sicily, Italy). We demonstrate that a blind filtering procedure can not properly account for local sources and would lead to a non representative selection of the important scenarios. For the specific source-target configuration, this results into an overestimation of the tsunami hazard, which turns out to be correlated to dominant coastal uplift. Different settings could produce either the opposite or a mixed behavior along the coastline. However, we show that the effects of the coseismic deformation due to local sources can not be neglected and a suitable correction has to be employed when assessing local scale SPTHA, irrespectively of the specific sign of coastal displacement.

## 1 Introduction

In the last fifteen years, a number of large earthquakes, often accompanied by destructive tsunamis, occurred worldwide. In several cases, the overall size of the earthquake and/or of the tsunami was unanticipated and some surprising features were observed, in terms of event scaling (e.g., source aspect ratio, tsunami height versus earthquake magnitude) or associated damage (Lay, 2015; Lorito et al., 2016); a striking example is the 2011 Tohoku earthquake and tsunami and the consequent

nuclear disaster at the Fukushima Dai-ichi power plant (Synolakis and Kânoğlu, 2015). These events called the attention for a systematic re-evaluation of current tsunami hazard estimates.

In the past, tsunami hazard was mostly studied through simulations of one or several scenarios, either "worst credible" (e.g., Tinti and Armigliato, 2003; Lorito et al., 2008; Tonini et al., 2011; Løvholt et al., 2012a) or representative for different selected return periods (e.g., Løvholt et al., 2006; Harbitz et al., 2012; Brizuela et al., 2014; Gailler et al., 2015). Such an approach can be useful either as a first screening of tsunami hazard or to realize very detailed assessments to inform emergency managers on the potential impact of specific scenarios. Traditionally, the latter is often done also as a result of probabilistic hazard disaggregation (Bazzurro and Cornell, 1999).

To account for the tsunami potential variability and frequency, and for including alternative models needed for quantifying the epistemic uncertainty, the probabilistic treatment of a large set of potential tsunami sources is essential. Probabilistic Tsunami Hazard Analysis (PTHA) probably begun with the seminal papers of Lin and Tung (1982) and Rikitake and Aida (1988). Uncertainty quantification is one of the main goals of PTHA, and progressively more refined uncertainty treatment was achieved following the 2004 Indian Ocean tsunami (e.g., Geist and Parsons, 2006; Burbidge et al., 2008; González et al., 2009; Horspool et al., 2014; Hoechner et al., 2016; Selva et al., 2016; Davies et al., 2017; Grezio et al., 2017; Power et al., 2017). PTHA is becoming the established good practice to manage risk assessment and risk mitigation measures (Chock et al., 2016; Løvholt et al., 2017). Due to the lack of historical tsunami data, the opportunity to deal with PTHA through a computational approach, involving the probability of all of the relevant sources and the numerical modeling of the generated tsunamis, which is in the scope of all the above mentioned papers, is emphasized for example by several reviews (Geist and Lynett, 2014; Grezio et al., 2017).

Nevertheless, the computational procedure for a complete evaluation of PTHA, fully honoring the natural variability of the sources, can be extremely demanding and unfeasible in some cases, particularly when inundation calculations are involved for a target site (González et al., 2009; Geist and Lynett, 2014). This is due to the very large number of numerical simulations of tsunami generation, propagation and inundation on high resolution topo-bathymetric models which is, in principle, required. For example, numerous realizations of heterogeneous slip are needed and usually obtained with stochastic procedures (LeVeque et al., 2016; Sepúlveda et al., 2017). Indeed, heterogeneous earthquake slip is known to strongly influence the tsunami run-up (Geist, 2002; Løvholt et al., 2012b; Geist and Oglesby, 2014; Davies et al., 2015; Murphy et al., 2016), not only in the the near-field of the source (Li et al., 2016). Among some first attempts towards quantifying tsunami hazard uncertainty related to heterogeneous earthquake slip, Mueller et al. (2014) and Griffin et al. (2017) should be mentioned. Recently, Goda and De Risi (2018) proposed a multi-hazard approach including stochastic slip distributions and cascading earthquake-tsunami risk evaluation; however they considered a limited number of tsunami scenarios, without fully characterizing the epistemic uncertainties associated with the key model components. Consequently, an efficient methodology is needed to make (onshore) PTHA a computationally affordable task.

The issue has been dealt with in various ways in several studies (González et al., 2009; Thio et al., 2010; Lorito et al., 2015; Lynett et al., 2016). In particular, Lorito et al. (2015) focused on Seismic PTHA (SPTHA), that is on hazard associated to tsunamis generated by coseismic seafloor displacement. They developed a method for significantly reducing the computational

cost of the assessment, by means of a source filtering procedure based on a cluster analysis. This allows the identification of a subset of important sources able to preserve the accuracy of results. Furthermore, Selva et al. (2016) proposed a general procedure for the joint and unbiased quantification of the aleatory and epistemic uncertainty, including the filtering procedure by Lorito et al. (2015) while stressing the importance of source completeness.

Here, we merge the two approaches by Lorito et al. (2015) and Selva et al. (2016), fully developing a method that enables the quantification of the local scale SPTHA, also devoting a big effort in refining the procedure and introducing several critical improvements. On one hand, we modified the filtering procedure to enhance its computational efficiency and to adapt it to multiple sources covering a large range of source-target distances. On the other hand, to improve the accuracy, we applied a separate treatment for remote and local sources, selecting near-field scenarios on the basis of the similarity of the coseismic

tsunami initial conditions. This is crucial, as near-field sources may challenge the general assumption made by Lorito et al. (2015), where, for a given source, offshore tsunami amplitude profiles are considered representative of the coastal inundation behind, regardless of the source location with respect to the coast. It was reasonable in that case study, since they considered either far-field scenarios with respect to the target coast or scenarios which deformed the coast in a definite direction, that is the coast always subsided due to subduction earthquakes on the Hellenic Arc. In the presence of more complex (and realistic)

local fault distribution, causing either subsidence or uplift or mixed patterns depending on the cases, tsunami intensity can be unpredictably reduced or enhanced with respect to the corresponding offshore tsunami wave (Mueller et al., 2014; Griffin et al., 2017). Hence, in general, offshore tsunami profiles could be strongly misleading when coseismic deformation of the coast occurs. This may in particular affect the tail of the hazard curves (i.e. largest intensities), to which local sources significantly contribute, as also demonstrated by the disaggregation analysis in Selva et al. (2016). For all these reasons, a special treatment

is needed for local sources, based on the source similarities, considering the coseismic onshore displacement, rather than the offshore tsunami wave similarity.

For illustrative purposes, we considered, as a use-case, a target site in the Central Mediterranean, that is the Milazzo oil refinery (Sicily, Italy), in the Southern Tyrrenian sea. This site was previously selected within the framework of the EU project STREST (http://www.strest-eu.org/) as a test case for multi-hazard stress test development for non-nuclear critical infrastruc-

tures.

It is worth noting that this paper is strictly methodological and is aimed to propose a computationally efficient procedure for local scale SPTHA, rather than providing a realistic site-specific hazard assessment. In fact, for the sake of simplicity and in order not to deflect the attention from the core of the method, no efforts have been dedicated to constrain and test the (regional) seismic rates, the local seismic sources and their geometry and dynamics, including slip distributions, as well as the accuracy of

topo-bathymetric data used in tsunami simulations. Moreover, the filtering procedure has been forced to minimize the number of explicit numerical simulations, allowing a relatively larger accepted error with respect to the complete initial set of sources due to the introduced approximations.

The paper is organized as follows: section 2 resumes the general outline of the method for SPTHA evaluation, as proposed by Lorito et al. (2015) and Selva et al. (2016), while the innovative developments are described in section 3; section 4 focuses

on the illustrative application; conclusive remarks are drawn in section 5.

## 2 A general review of the original method for SPTHA

Using regional scale SPTHA as input for local scale (site-specific) SPTHA, through the approach proposed by Lorito et al. (2015), is a task already foreseen by Selva et al. (2016) (see Fig. 1 therein). However, this possibility was neither applied nor tested in practice, since their main focus was the application to regional scale analyses. The details of the general method have been already thoroughly described and validated in the previous studies. Here we will summarize the basic concepts.

The whole general procedure for site specific SPTHA can be outlined in four steps: (1) the definition of earthquake scenarios and their probability, allowing in principle a full exploration of source aleatory uncertainty; (2) the computation, for each source, of tsunami propagation up to a given offshore isobath; (3) the selection of the relevant scenarios for a given site through a filtering procedure and the relative high-resolution tsunami inundation simulations; (4) the assessment of local SPTHA with joint aleatory and epistemic uncertainty quantification by means of ensemble modeling, including modeling alternatives eventually implemented at steps (1)-(3).

In step (1), all the modeled earthquakes must be defined for different seismic regions, which are assumed to be independent from each other. The earthquake parameters and their logically ordered conditional probabilities are treated by means of an event tree technique. We emphasize that the common assumption that tsunami hazard is dominated at all time scales by subduction zone earthquakes is not used: non-subduction faults, unknown offshore faults and diffuse seismicity around major known and well mapped structures are all taken into account. This strategy attempts to prevent biases in the hazard due to incompleteness of the source model (Basili et al., 2013; Selva et al., 2016). The seismicity related to the main and better known fault interfaces is treated separately from the rest of the crustal and diffuse seismicity. A similar approach has been used in the recent TSUMAPS-NEAM project (http://www.tsumaps-neam.eu), which provided the first SPTHA model for the North-Eastern Atlantic, Mediterranean and connected seas (NEAM) region.

In step (2), for each scenario retrieved from step (1), the corresponding tsunami generation and propagation is numerically modeled, and the pattern of offshore tsunami height above the sea level ($H_{max}$) is evaluated on a set of points along the $50m$ isobath, in front of the target area. To provide the input to Lorito et al. (2015), these points may be limited to a profile in front of the site. The length of this control profile must be tuned depending on the morphology and the extension of the target coast: a trade-off has to be reached, as few points could make the profile not representative enough, while too many points could downgrade the performances of the subsequent filtering procedure (Lorito et al., 2015). Actually, the optimal length is the shortest one that makes the offshore hazard curves stable with respect to the source selection, and further increase in length would increase the computational effort without significantly altering the results.

In step (3), using the offshore $H_{max}$ profiles calculated at step (2), a filtering procedure is implemented to select a subset of relevant sources, based on the similarity of the associated tsunami intensity, not on the similarity or spatial proximity of the sources themselves. The selected sources, each of them representative of a cluster of sources producing comparable tsunamis offshore the target area, are then used for explicit inundation modeling on high-resolution topo-bathymetric grids. This approach allows for a consistent and significant reduction of the computational cost, while preserving the accuracy. However, Lorito et al. (2015) considered a limited set of sources. The extension to a much larger set of potential sources

requires some modification to the method that, along with several other improvements, are proposed in this study, as reported in section 3.

Incidentally, we note that other wave properties such as period or polarity could be relevant in the framework of the cluster analysis. However, Lorito et al. (2015) briefly discussed this issue, also with respect to the length of the control profile as discussed above. Nevertheless, this is a point probably deserving further investigation, considering that Satake et al. (2013) showed how inundation from the Tohoku 2011 tsunami was variably controlled by long-period offshore tsunami components on flat coastal plains and shorter-period peaks in steep coastal areas. Indeed, Gusman et al. (2014) used two cycles of a tsunami for identifying similar waves. Conversely, since, as described in the next section, offshore wave comparison is not here used anymore in the near-field, this issue will not apply for local sources.

In step (4), local SPTHA is quantified. The inundation maps for each representative scenario from step (3) are aggregated according to the probabilities provided at step (1), assigning the total probability of a cluster to the representative scenario. Aleatory and epistemic uncertainty are simultaneously quantified by means of an ensemble modeling approach (Marzocchi et al., 2015; Selva et al., 2016) over alternative implementations of the previous steps. In practice, steps (1) to (3) can be iterated for each alternative model and these alternatives can be weighted according to their credibility and the possible correlations among the models. The results are finally integrated through ensemble modeling into a single model which expresses both aleatory and epistemic uncertainty.

## 3   Improvements in the filtering procedure

The described method has been tested by both Selva et al. (2016) and Lorito et al. (2015). However, Lorito et al. (2015) focused on the filtering procedure of step (3), adopting a simplified configuration for the source variability, in which sources were allowed only within the Hellenic arc, that is an area relatively smaller than the full aleatory variability. On the other hand, Selva et al. (2016) applied the approach to a regional study extended to the Ionian Sea, in Central Mediterranean. The quantification of the local hazard is instead discussed only in theory, without proposing any application.

The original method by Lorito et al. (2015) adopted a two-stage procedure.

In the first stage, scenarios giving a negligible contribution to $H_{max}$ offshore the target area were removed, assuming they would lead to negligible inundation. Hereinafter, we call this stage "Filter H".

As a second filtering stage, a Hierarchical Cluster Analysis (HCA) was carried out, separately for each earthquake magnitude class included in the seismicity model, under the assumption that sources producing similar offshore $H_{max}$ along the control profile will also produce similar inundation patterns. The distance between two $H_{max}$ patterns from two different scenarios $u$ and $v$ was measured by a cost function previously used to compare tsunami waveforms in source inversion studies (e.g., Lorito et al., 2010; Romano et al., 2010) and modified by Lorito et al. (2015) as

$$d(H_{max}^u, H_{max}^v) = \left[1 - \frac{2\sum_x H_{max}^{u,x} H_{max}^{v,x}}{\sum_x (H_{max}^{u,x})^2 + (H_{max}^{v,x})^2}\right], \tag{1}$$

where $x$ runs over the control points on the $50m$ isobath. For each cluster, the scenario closer to the centroid was selected as the reference scenario, with an associated probability corresponding to the probability of occurrence of the entire cluster. The optimal number of clusters (i.e., the "stopping criterion") was assessed by analyzing the variance within each cluster (hereinafter "intra-cluster") as a function of the number of clusters and selecting the largest value still producing significant

changes, according to the so-called Beale test (Lorito et al., 2015, and references therein).

We implemented a different strategy to further reduce the number of explicit tsunami simulations and introduced a separate treatment for local and remote sources. In particular, the source scenario filtering procedure was revised to improve both the computational efficiency and the accuracy, allowing for a full scalability to the source variability of typical SPTHA (millions of scenarios located allover an entire basin). A schematic diagram of the new procedure is sketched in Fig. 1, with (right, step

(3b)) or without (left, step (3a)) the separation between near- and far-field.

We still kept Filter H, but also adopted an additional filter on the occurrence probability (hereinafter "Filter P", see Fig. 1), discarding scenarios whose cumulative mean annual rate (mean of the model epistemic uncertainty) is below a fixed threshold. Filter P works as follows. Scenarios are sorted according to their mean annual rate and the rarer are removed until the cumulated rate reaches the selected threshold. This allows to further reduce the number of required numerical simulations. On

the other hand, this operation introduces a controlled downward bias on the estimated hazard, whose upper limit corresponds (on average) to the probability threshold of the filter P. This threshold can be set at a negligible level in the framework of the overall analysis and/or with respect to other uncertainties. In addition, it can be empirically checked to which extent this affects the results by analyzing the offshore hazard curves at the control points. This check was quantitatively done by computing the maximum deviation between the mean hazard curves at each control point before and after Filter P was applied. We also

notice that, as reported in Fig. 1, Filter P was always applied after Filter H due to strategical reasons of optimization: in fact, the cumulate rate curve is lowered by the removal of small events (i.e., producing small $H_{max}$), which are typically featured by high occurrence probability. As a consequence, a greater number of scenarios can be removed before reaching the imposed threshold, making Filter P more efficient.

Also the cluster analysis stage was modified. Firstly, we used a different algorithm, as the large number of source scenarios,

due to a realistic fault variability distribution, in some cases can make the HCA a computationally unaffordable task. We implemented the more efficient $k - medoids$ clustering procedure (Kaufman and Rousseeuw, 2009; Park and Jun, 2009), based on the minimization of the sum of the intra-cluster distances, that is the distances between each element of a cluster and the cluster centroid. Strong constraints on the distances result in a more accurate partitioning, in terms of similarity among the elements of each cluster, but lead to a great number of clusters. Instead, larger ranges of acceptability increase the efficiency

of the algorithm, in terms of number of resulting clusters, to the detriment of the accuracy. The cluster analysis was performed separately for groups of scenarios with similar mean $< H_{max} >$ along the profile, instead of grouping scenarios per earthquake magnitude classes. This makes the partitioning more efficient, as the earthquake magnitude can not be considered the only parameter controlling the tsunami intensity, as it was for the limited set of sources adopted by Lorito et al. (2015). The cluster distance was measured by the eq. 1, but we updated the stopping criterion, which is now related to the maximum allowed intra-

cluster variance, rather than being a blind optimization of the number of clusters. More specifically, to control the dispersion

within each cluster, we set a threshold for the maximum allowed squared Euclidean distance. This threshold was empirically fixed by comparing the offshore hazard curves before and after the analysis and assuming an acceptable range of variability, in analogy with the approach used for Filter P.

Finally, and probably most importantly, in order to deal more properly with the contribution from local sources, we implemented two independent filtering schemes for distant and local sources. Indeed, a special treatment for near-field sources is needed, as the coseismic deformation can modify the actual local tsunami intensity at the nearby coast, due to coastal uplift or subsidence. As a consequence, the offshore tsunami amplitude profiles generated by such events may fail in being representative of the coastal inundation, and a separate modeling is required, using the coseismic deformation as the metric for source proximity in the cluster analysis (details below). This issue was somehow hidden in Lorito et al. (2015), due to the relatively small aleatory variability they considered, being the source either in the far- or near-field, depending on the target site, but never mixed together. In addition, this separation may favor some refinement of the near-field source discretization and modeling, such as a denser sampling of geometrical parameters and/or the introduction of heterogeneous slip distributions.

For testing the proposed method, we replaced step (3) either with step (3a) or step (3b), as displayed in Fig. 1. The workflow of step (3a) is substantially equivalent to the original procedure by Lorito et al. (2015), improved by the aforementioned changes related to the algorithm optimization, whereas the separate treatment of near- and far-field sources is included in step (3b). Step (3a) is then used in this study as a term of comparison for the new scheme.

In step (3a), three sequential tasks were performed, namely Filter H, Filter P and the cluster analysis based on the offshore tsunami amplitudes.

In step (3b), local and distant sources were firstly detected, based on the coseismic deformation produced by the earthquake near and on the target coast. The procedure was then split into two parallel paths, which need to be merged at the end when evaluating SPTHA (Fig. 1). As far as the far-field scenarios are concerned, the same workflow as step (3a) was followed. Near-field scenarios, which in principle should be individually modeled, were also filtered in order to reduce the number of explicit inundation simulations: this of course introduces a new approximation, which however is better than aggregating local and remote scenarios on the basis of the offshore tsunami amplitudes. Filter H was applied as well, but choosing a smaller threshold value: a more conservative approach is indeed recommended at this stage, as offshore values could be strongly misleading when significant coastal coseismic deformation occurs. Then, Filter P was employed and finally a cluster analysis was performed, by comparing the coseismic deformations, instead of the (unrepresentative) offshore tsunami amplitudes. For each local source, the vertical component of the coseismic displacement was calculated on a 2D grid centered around the fault and having size equal to three times the fault length. Then, the cluster analysis was carried out, separately for each magnitude, by comparing the coseismic fields point-to-point within the grid. In this case, the cluster analysis is based on the squared Euclidean distance, instead of the cost function; also the stopping criterion is evaluated through the Euclidean distance, since the coseismic field can take both positive and negative values.

The selected earthquake scenarios from step (3a) or from the two branches (near- and far-field) of step (3b) were then used for high-resolution inundation simulations and combined together in step (4) when evaluating SPTHA. A practical example of the whole procedure is illustrated in the next section.

## 4 The Milazzo oil refinery (Sicily, Italy) use-case

The described procedure was applied to a test site, Milazzo, located on the north eastern coast of Sicily (Italy), within the Mediterranean Sea. The site houses an oil refinery, one of the non nuclear critical infrastructures selected as case study in the framework of the EU project STREST (http://www.strest-eu.org/).

Due to the illustrative purposes of the present work, some strong assumptions were imposed during the filtering procedure to drastically reduce the number of required explicit numerical simulations. The tuning of the filtering thresholds is not the objective of the present work; in fact, the application is aimed to highlight that inaccurate (biased) evaluation of site-specific tsunami hazard would be obtained if scenarios located in the near-field of the target area are not properly taken into account, irrespectively of the completeness and consequent complexity of the hazard assessment. However, more sanity and sensitivity tests for a finer tuning of thresholds and modeling would be mandatory in case of a real application. For example, the modeling of near-field scenarios is expected to be dependent on the source parameters, especially concerning the heterogeneous slip distribution on the fault plane (e.g., Geist and Oglesby, 2014), which was not included here. Hence, the computational effort of a real assessment, including a wider source variability and more conservative thresholds, is expected to be more complicated and computationally demanding than this case-study.

Regarding step (1), the adopted seismicity model was previously developed in the framework of the EU project ASTARTE (http://www.astarte-project.eu/). This model extends the method applied to the Ionian Sea in Selva et al. (2016) to the entire Mediterranean Sea, including the subduction interfaces of the Calabrian and Hellenic Arcs as well as crustal seismicity in the whole basin (see Fig. 2a). On subduction zones, events of different magnitude and positions on the whole interface are allowed, disregarding the geometry uncertainty of the slab; conversely, crustal seismicity is allowed to occur with any meaningful geometry and mechanism in the whole seismogenic volume at different magnitude and depths. The complete set of sources retrieved from step (1) contains about 40 millions of elements, among which 1,701,341 scenarios actually affect the target site ($H_{max} > 0.05m$ offshore Milazzo). Although relatively simplified, the source model includes also epistemic uncertainties on many source parameters such as the seismic rates, the shape of the magnitude-frequency distribution, and the seismogenic depth interval for the two subduction zones.

Tsunami amplitudes (step (2)) were computed on a control profile made of 11 points offshore the Milazzo target area (on the $50m$ isobath), as reported in Fig. 2A. To save computational time, scenarios from step (1) were not individually simulated, but were obtained by linear combination of pre-calculated tsunami waveforms produced by Gaussian-shaped unitary sources (Molinari et al., 2016). The Gaussian propagation has been modeled by the Tsunami-HySEA code, a non-linear hydrostatic shallow-water multi-GPU code based on a mixed finite difference/finite volume method (de la Asunción et al., 2013; Macías et al., 2016, 2017).

Step (3) was addressed by independently performing the two branches (3a) and (3b), as discussed in the previous section, and then comparing results to assess the importance of the separate treatment of the near-field sources.

In step (3a), thresholds were fixed at $1m$ for Filter H and $10^{-5}yr^{-1}$ for Filter P. This resulted in discarding scenarios with individual mean annual rate below $\sim 10^{-9}yr^{-1}$, causing a maximum bias on the offshore mean hazard curves of about

10% in the considered range of tsunami intensities, with respect to the curves obtained without Filter P. At the end of the filtering procedure, imposing a threshold equal to 0.2 on the intra-cluster variance, we obtained 776 clusters, each associated to a representative scenario. That is, we had a reduction above 99%. Figure S1 of the Supplementary Material shows the comparison among the mean offshore hazard curves at the 11 control points, as well as among some quantiles of the epistemic uncertainty, for the filtered and original set of scenarios.

It is worth stressing that the efficiency of the filters is here artificially enhanced by the imposed high thresholds, especially as far as the Filter H is concerned. While $1m$ is not an acceptable value in case of a real hazard assessment, it is suitable for illustrative purposes. In any case, this filter, independently from the chosen threshold, is not expected to affect subsequent steps of the procedure for tsunami intensities above the threshold. Conversely, we performed a sensitivity analysis on the threshold imposed on the intra-cluster variance for the cluster analysis: Fig. S2 shows the percentage differences between the offshore hazard curves computed from the complete initial set of sources and the filtered set. The red box corresponds to the chosen threshold value (0.2): it appears evident that a smaller value would have allowed a stronger constraint on the error introduced by the cluster analysis, while considerably increasing the number of resulting clusters. Vice versa, higher thresholds produce a smaller number of clusters, but fail in reproducing the hazard (error up to 40%). In case of a real hazard assessment, this analysis would help choosing an optimal threshold.

In step (3b), we considered as local scenarios, requiring a separate processing, sources generating a coseismic vertical displacement greater than or equal to $0.5m$ on a set of near-field points, that is the 11 control points on the $50m$ isobath plus 95 inland points, strategically located at the edges of the refinery storage tanks, as shown in Fig. 2B. We found 4721 scenarios in the near-field (see Fig. 2A). Afterward, for both branches we applied Filter H and P as well, using the following thresholds: for far-field scenarios, Filter H=$1m$ and Filter P=$5 \times 10^{-6} yr^{-1}$; for near-field scenarios, Filter H=$0.1m$, according to the more conservative approach described in the previous section, and Filter P=$5 \times 10^{-6} yr^{-1}$. Note that Filter P threshold was set half the value used in step (3a), in order to keep a total maximum theoretical bias on the hazard curves at $10^{-5} yr^{-1}$ (as in step (3a)), considering that Filter P is separately applied both to far- and near-field scenarios. Then, the cluster analysis was carried out on the tsunami amplitudes for far-field scenarios (using a threshold equal to 0.2 on the intra-cluster variance) and on the coseismic deformation for near-field scenarios (using a 10% threshold for the intra-cluster variance). We obtained 634 and 520 clusters for remote and local sources, respectively. Thus, the total number of representative scenarios (1154) to be explicitly modeled corresponds to a reduction above 99% of the initial set of sources.

Inundation simulations at step (3) have been carried out again with the Tsunami-HySEA code, exploiting the nested grid algorithm. We used 4-level nested bathymetric grids with refinement ratio equal to 4 and increasing resolution from $0.4 arc - min$ ($\sim 740m$) to $0.1 arc - min$ ($\sim 185m$) to $0.025 arc - min$ ($\sim 46m$) to $0.00625 arc - min$ ($\sim 11m$). The largest grid was obtained by resampling the SRTM15+ bathymetric model (http://topex.ucsd.edu/WWW_html/srtm30_plus.html). The finest three grids have been produced by interpolation from TINITALY (inland, Tarquini et al. (2007, 2012)) and EMODNET (offshore, http://www.emodnet-bathymetry.eu/), working on grids of $0.00625 arc - min$ that have been resampled at $0.1 arc - min$ and $0.025 arc - min$. A picture of the telescopic nested grids is provided in Fig. S3 of the Supplementary Material. The initial conditions were differently provided for subduction and crustal seismicity. The subduction scenarios have been simulated by

modeling the slab as a 3D triangular mesh honoring the interface profile and using unitary Okada sources associated to each element of the mesh (i.e., to each triangle) as Green's functions (Okada, 1985; Meade, 2007). For crustal events, the initial sea level elevation was obtained by modeling the dislocation on rectangular faults according to the Okada model. A Kajiura-like filter for the sea-bottom/water-surface transfer of the dislocation was also applied (Kajiura, 1963). For each simulation an overall length of 8 hours was fixed. The results were stored as maximum wave height ($H_{max}$, m) and maximum momentum flux ($MF_{max}$, $m^3 s^{-2}$), at each point of the inner grid.

At step (4), SPTHA was evaluated in parallel using results both from steps (3a) and (3b), in order to compare the outcomes of the two different workflows and estimate the impact of the special treatment of near-field sources on the site-specific hazard assessment. Note that alternative models for the epistemic uncertainty were considered only at step (1), that is only as far as the probabilistic earthquake model is concerned, since the Selva et al. (2016) model was used.

Figures 3 to 5 compare the results from steps (3a) and (3b), in terms of mean hazard curves and inundation (both probability and hazard) maps for $H_{max}$. At a first glance, differences are appreciable in both the curves and the maps. It is worth noting that results at $H_{max} < 1m$ can be (negatively) biased since they are depleted from the scenarios removed by Filter H , both in step (3a), as clearly shown in Fig. S1, and in the far-field-branch of step (3b). Curves and maps will be described in more detail in the following.

The hazard curves in Fig. 3 (panels a) and b)) show the mean (mean of the model epistemic uncertainty) exceedance probability in $50yr$ for $H_{max}$ (evaluated assuming a Poisson process, as in Selva et al. (2016)), plotted for each point of the finest resolution grid. Panel c) of the same figure displays the one-by-one relative differences in terms of exceedance probability (in $50yr$), as a function of $H_{max}$, between the step (3a) and (3b) curves at each grid point. For values of $H_{max}$ greater than $1m$, the relative differences are systematically positive, meaning that without the correction for near-field scenarios (step (3a)), the tsunami hazard would be overestimated. In Fig. S4 of the Supplementary Material a sample of curves at few inland points (one every thousandth) is displayed for a direct curve-by-curve comparison between the two approaches. This confirms that, overall, the uncorrected approach leads to hazard overestimation. We may argue that this is true in the case of this specific setting, as a lower "corrected" hazard means that the predominant effect by local sources contributing to a specific point on the hazard curve is due to the coastal uplift, which in turn decreases the tsunami hazard. For example, a cluster may mix far- and near-field sources, which could be misrepresented by one far-field source selected as cluster representative. In our case, there might be a prevalence of clusters causing coastal uplift from the near-field sources. The situation may be the opposite for a different source-target configuration, that is coastal subsidence could be predominant causing an hazard increase, which without correction would be underestimated. To confirm our inference, we performed some further testing. For each hazard intensity, and only for the mean model of the epistemic uncertainty, we computed the coseismic coastal displacement in the inner grid, averaged both over all of the scenarios and over all of the coastal points (purple line in Fig. 3c). This quantity can be regarded as the mean uplift (hereinafter MU) on a random point on the coastline. Scenarios of different types contribute to MU, both far-field scenarios, which do not alter the coastline, and near-field scenarios, which may include a mixture of sources producing both coastal subsidence and uplift. More in detail, we firstly performed, for each $H_{max}$, a weighted average of the coseismic displacements from each cluster centroid, with weights equal to the annual probability of the individual earthquakes.

These probabilities are set to zero if the earthquake do not deform the coastline (i.e. for far-field sources) or if the generated tsunami does not exceed the given $H_{max}$ value (i.e. that scenario does not contribute to the hazard at that point). The weighted average is then normalized to the total probability of the near- and far-field sources contributing to the tsunami hazard for that threshold. The resulting MU on each coastal point is plotted, for different values of $H_{max} \geq 1m$, in Fig. S5 of the Supplementary Material (blue lines). The displacements due to the single cluster representatives are also shown (red lines). We note that (i) although single scenarios produce both positive and negative coastal displacements, the predominant contribution is unveiled by the sum over the different clusters, which is definitely positive; (ii) for the higher intensities, the contributing scenarios (decreasing in number, as expected) generate displacements which are smaller and smaller, as important uplift would significantly limit tsunami inundation. Finally, we further averaged the resulting values along the coastline, obtaining the purple curve in Fig. 3c. We notice that the absolute MU value in meters turns out to be rather small, as a result of the average over sources that cause either uplift or subsidence, or no coastal displacement at all. Anyway, the positive values obtained for $H_{max} > 1m$ indicate that the uplift of the coast is prevailing, consistently with the positive percentage differences retrieved between the two approaches. Very little differences are retrieved between the "corrected" and the "uncorrected" filtering procedures for smaller values of $H_{max}$, that is below the Filter H threshold.

Finally, in Fig. 3d the relative differences are also shown in terms of $H_{max}$ as a function of exceedance probability (in $50yr$). In the low probability region, typically corresponding to high $H_{max}$, the overestimation by step (3a) is confirmed; conversely, for exceedance probability greater than $\sim 10^{-4}$, which is likely to correspond to small $H_{max}$, a greater dispersion with both positive and negative values is observed.

Probability and hazard inundation maps can be achieved by vertically and horizontally cutting the hazard curves at chosen fixed values, in order to give a geographical representation of results. As each hazard curve corresponds to a grid point, the probability maps are obtained by plotting on a map all the probability values for a fixed value of the intensity metric. Instead, in the hazard maps the intensity values are plotted for a fixed exceedance probability, corresponding to a given ARP. In Fig. 4 we computed the exceedance probability maps for $H_{max} = 2m$ and $H_{max} = 3m$, while in Fig. 5 we extracted the hazard maps for $ARP = 2 \times 10^5 yr$ and $ARP = 3 \times 10^5 yr$ (corresponding to an exceedance probability in $50yr$ equal to $2.5 \times 10^{-4}$ and $1.7 \times 10^{-4}$, respectively).

For the selected values, the maps confirm what we already discussed about the curves: from the probability maps, mostly positive relative differences both inland and offshore are inferred, as shown in panels (c) and (f) of Fig. 4, even larger than 50%; these differences are positive in a larger number of points for the higher intensity, consistently with Fig. 3c. We recall that positive differences mean that the "uncorrected" procedure (step (3a)) actually overestimates the tsunami hazard at the target site. Negative inland values are also observed for $H_{max} = 2m$, but they occur for very low probability values and should not be further investigated. We also notice that the area inundated with a non negligible probability decreases in size with increasing the $H_{max}$ value, as expected. In the hazard maps (Fig. 5) a complex pattern is revealed when inspecting the relative differences (panels (c) and (f)), as both positive and negative values are retrieved. This happens because the analyzed ARPs lie in the low intensity range. The inundated area, as opposite to the previous case, is consistently more extended for larger ARPS.

Further details about the comparison can be found by analyzing the curves and the maps for $MF_{max}$ reported in Figs. S6 to S8 of the Supplementary Material. We just note that, when the correction for near-field is taken into account, the inundation maps highlight an enhanced current vorticity near the docks (Fig. S7(b,e) and S8(b,e)), which is a known effect due to the flow separation at the tip of a breakwater (Borrero et al., 2015). As the probability and hazard maps aggregate several different sources, the hazard integral may tend to average and cancel out different source effects, while enhancing local propagation features. The presence of such persistent physically meaningful effects only in the maps retrieved using step (3b) confirms the importance of the special treatment. In other words, the blind cluster analysis (step (3a)), exclusively based on the offshore tsunami amplitudes, likely produced a non-representative selection of the important scenarios, as it could aggregate or even remove important local scenarios.

## 5 Conclusions

We proposed a computationally efficient approach to achieve robust assessment of site-specific SPTHA, developing an improved version of the method by Lorito et al. (2015) and Selva et al. (2016).

The procedure is based on 4 steps, which can be summarized as follows:(1) the definition of the set of earthquake scenarios and their mean annual rates, exploring the source aleatory uncertainty; (2) the computation of tsunami propagation up to an offshore isobath; (3) the implementation of a filtering procedure to select relevant scenarios for the target site, which are then explicitly modeled; (4) the assessment of local SPTHA through an ensemble modeling approach, to jointly quantify aleatory and epistemic uncertainty, stemming from alternative models for steps (1)-(3).

In the present work we focused on step (3), modifying the filtering procedure to enhance the computational efficiency and introducing a separate treatment for sources located in the near-field, to take into account the effect of the coseismic deformation on the tsunami intensity. To this aim, we implemented a new procedure including a correction for near-field and some numerical improvements. We benchmarked the new approach against an algorithm essentially equivalent to the original method by Lorito et al. (2015). The correction is crucial as the latter is based on the assumption that offshore tsunami profile is representative of the inundation at the nearby coast, which might be true if a coseismic deformation of the coast is not involved; otherwise seafloor uplift or subsidence make the assumption invalid. Consequently, local and remote sources must be separately treated by means of different filtering procedures. This may also allow for a specific and more detailed parameterization of the near-field sources, to which the local hazard is known to be more sensitive.

We tested the procedure investigating a case study, i.e. Milazzo (Sicily). The work has only illustrative purposes and is not intended as a real hazard assessment at that site, due to some simplifications in the adopted model.

The new implemented filtering procedure allows for a consistent reduction of the number of tsunami inundation simulations and therefore of the computational cost of the analysis. It is worth stressing that in this specific application the computational efficiency was artificially enhanced by limiting the source variability as well as by imposing high filter thresholds. In fact, a real assessment is expected to deal with a greater number of scenarios, provided that a finer tuning of the threshold values is

carried out. This may in particular affect the computational cost related to the analysis of the near-field sources, for example when using stochastic slip distributions.

The most striking result is that the separate treatment of near-field sources provides significantly different and physically more consistent results with respect to the "uncorrected" procedure, showing that near-field sources must be specifically dealt with when evaluating site-specific SPTHA. We recall that the two approaches (with or without the correction for near-field) only differ in the way local sources are treated. Hence, the different results do not depend on the specific filtering thresholds but just on the coseismic deformation induced by local sources, which, if properly accounted for, modifies the effective tsunami hazard. Actually, for the specific configuration of this use-case, our findings reveal that not considering an appropriate correction for near-field would lead to overestimate the tsunami hazard for $H_{max}$ greater than $1m$, and this overestimation is correlated to dominant coastal uplift. However, different cases in terms of over- and under-estimation may occur at different sites, depending on the relative source-site configuration. We also observe that Milazzo is located in an area featuring relatively low near-field tsunamigenic seismicity with respect to other areas in the Mediterranean sea. Nevertheless, the method turns out to be sensitive even to relatively low displacements and allows to detect and remove significant biases from near-field sources.

The proposed method is suitable to be applied to operational assessments, also for improving local (multi-hazard) risk analyses (e.g. Goda and De Risi, 2018). We stress again that the approach developed here allows to consider a very high number of tsunami scenarios, which is necessary to sufficiently explore the natural variability of the tsunami sources and the eventual alternative models needed for quantifying the epistemic uncertainty.

Future work will be devoted to use the procedure to perform real local hazard assessment, exploiting the regional hazard retrieved from the TSUMAPS-NEAM project.

*Competing interests.* The authors declare that they have no conflict of interest.

*Acknowledgements.* The authors want to thank the EDANYA Research Group at University of Malaga for providing the Tsunami-HySEA code for tsunami simulations. We acknowledge useful discussions with William Power and Gareth Davies during the early stages of this work. We also acknowledge constructive comments by three anonymous referees, which allowed a significant improvement of this paper. The work was partially funded by INGV–DPC Agreement (Annex B2) and by the STREST project, EC's Seventh Framework Programme [FP7/2007-2013], grant agreement n. 603389. All of the figures have been created using either MATLAB (www.mathworks.com) and Generic Mapping Tools (http://gmt.soest.hawaii.edu).

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

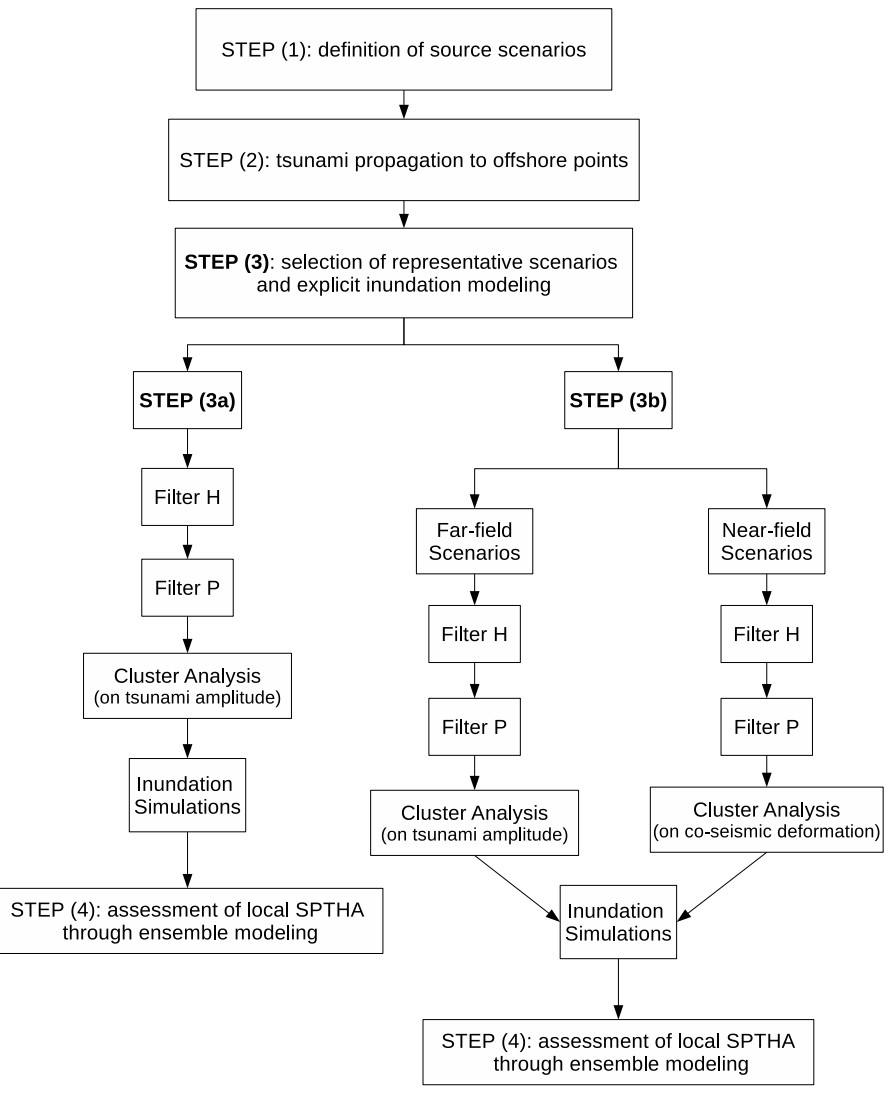

**Figure 1.** Schematic diagram of the computational procedure to evaluate site-specific SPTHA, with special attention to step (3) (see text).

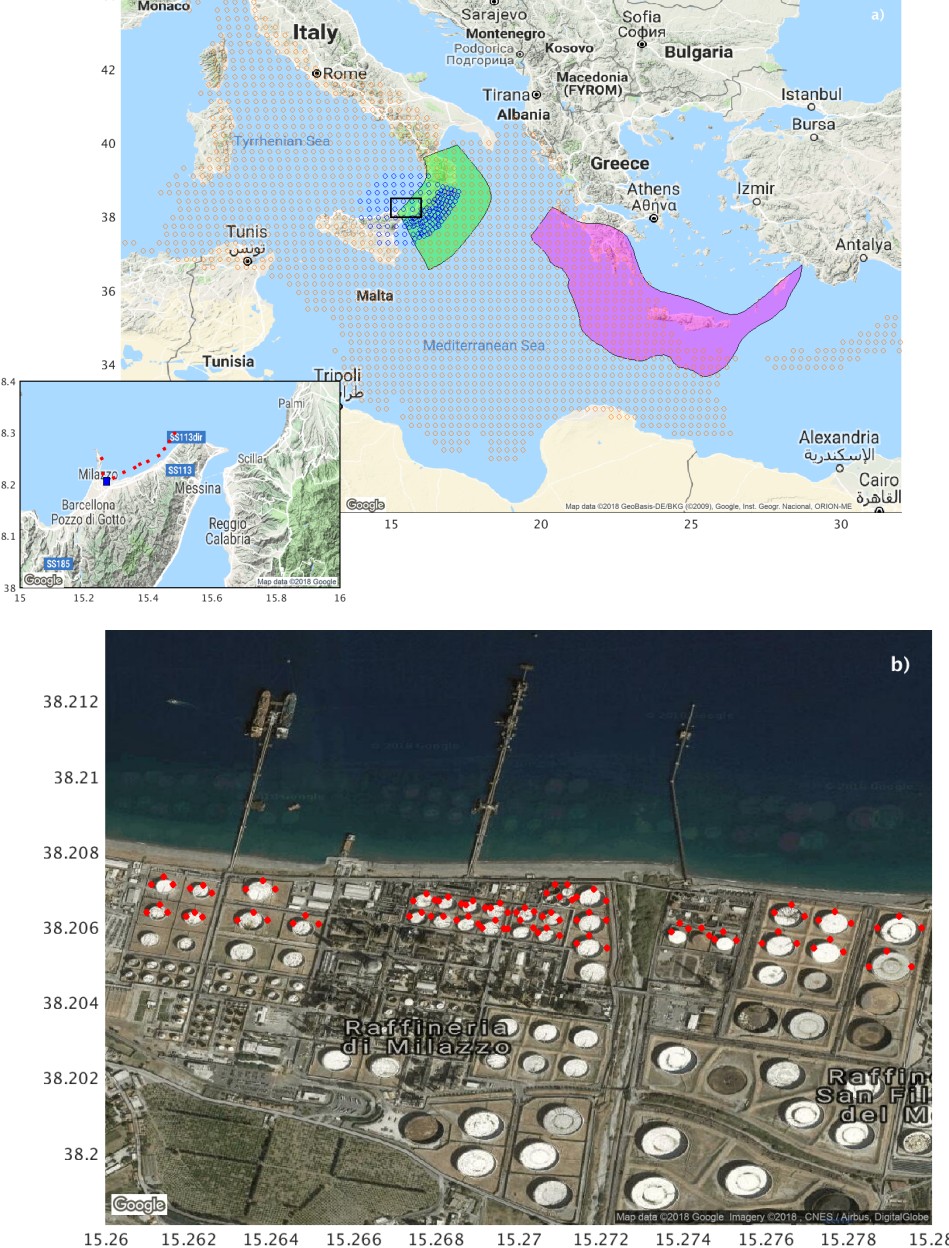

**Figure 2.** a) Map of the whole simulation domain used for the application at the target site Milazzo (Sicily, Italy). The orange circles are the geometrical centers of the crustal faults affecting the target site, while the magenta and the green regions are the slab models of the Hellenic and Calabrian arc respectively. Blue circles are the geometrical centers of the near-field sources, as detected in step (3b) (see text). The inset highlights the offshore points along the $50m$ isobath (red points). The blue rectangle within the zoom is the area displayed in the bottom panel. b) Zoom on the Milazzo oil refinery, with the position of the 95 points at the edges of the storage tanks (red points).

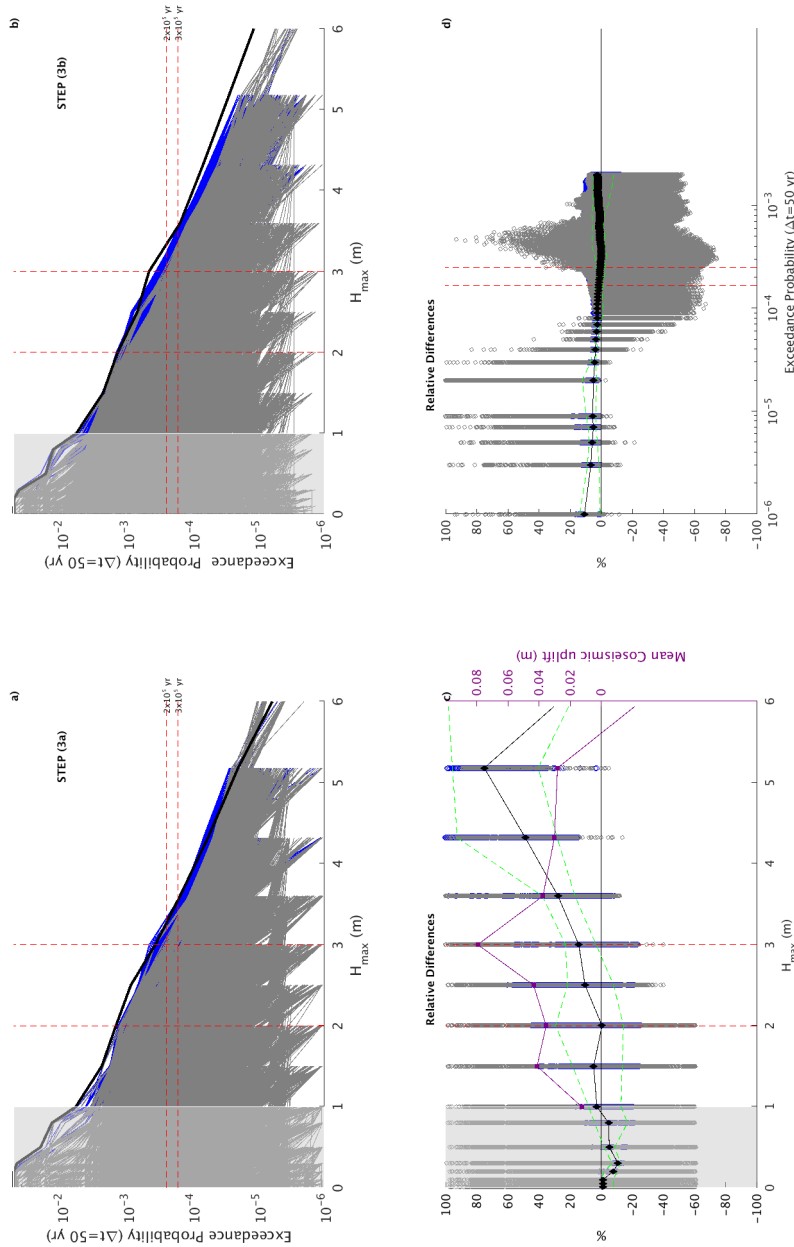

**Figure 3.** a) Mean hazard curves for $H_{max}$ at all points within the highest resolution grid, as obtained from step (3a) of the SPTHA procedure (see text and Fig. 1). Grey and blue colors refer to inland and offshore points, respectively. The bold black line represents the envelope of the curves from step (3b). Red dashed lines represent the values used to obtain probability (Fig. 4) and hazard (Fig. 5) inundation maps. b) Same as a) but using step (3b). The bold black line is the envelope of the curves from step (3a). c) Relative differences in terms of exceedance probability (in $50yr$) as a function of $H_{max}$, computed as $[(3a) - (3b)]/(3b)$. The black line is the median of the point distribution; the green dashed lines correspond to the $16^{th}$ and $84^{th}$ percentile. The MU, namely the mean uplift on a random point along the coastline (see text) is also superimposed (purple line).

d) Same as c) but in terms of $H_{max}$ as a function of the exceedance probability (in $50yr$).

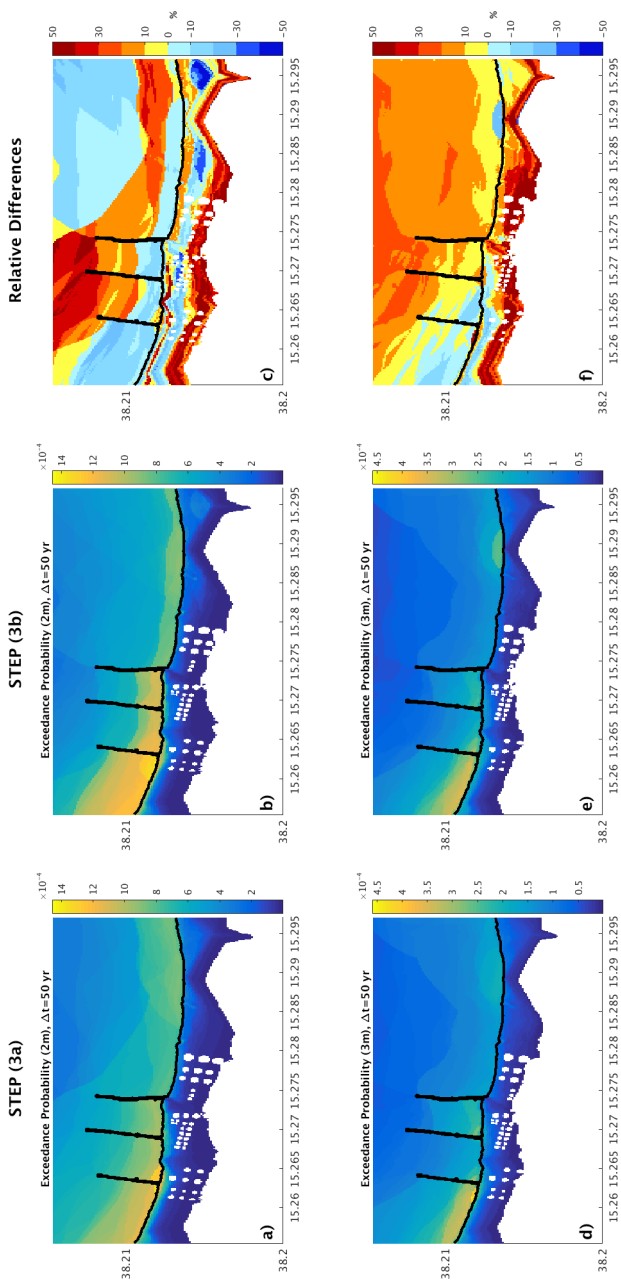

**Figure 4.** Probability maps (inner grid) for $H_{max}$ derived from the hazard curves in Fig. 3 at two different thresholds ($2m$, $3m$) for step (3a) and (3b) and relative differences computed as $[(3a) - (3b)]/(3b)$.

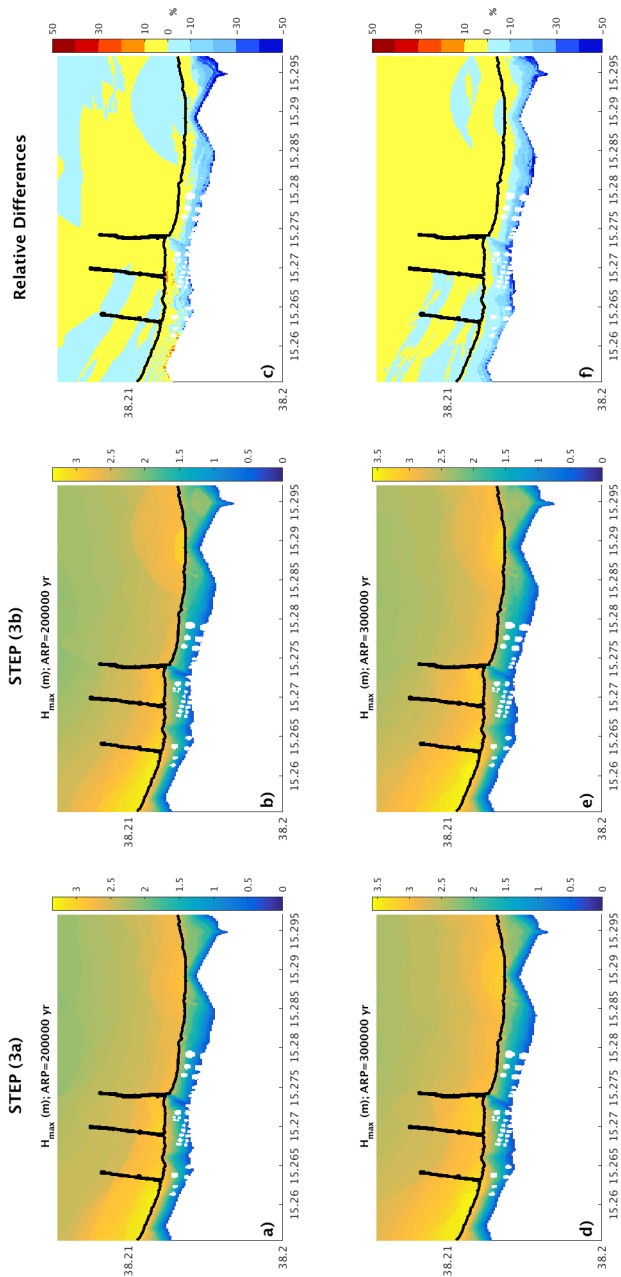

**Figure 5.** Hazard maps (inner grid) for $H_{max}$ derived from the hazard curves in Fig. 3 at two different ARPs ($2 \times 10^5 yr$, $3 \times 10^5 yr$) for step (3a) and (3b) and relative differences computed as $\left[(3a) - (3b)\right]/(3b)$.