# Peer review of "From regional to local SPTHA: efficient computation of probabilistic tsunami inundation maps addressing near-field sources"

_Natural Hazards and Earth System Sciences, 2018_

## Referee Comment (RC1) · Anonymous Referee #1 · 6 Aug 2018

**# # Overall comments**

This looks to be a good paper about an important topic that is of clear interest to NHESS readers. It is mostly well written, and describes innovative ideas which are likely to be of broad utility in tsunami hazard assessment.

My only 'significant' concern is that the authors do not provide a 'conceptual justification' for the differences in the results of the two filtering methods they apply, for the case of high $H_{max}$. As it stands, as a reader I don't know why this happens. Intuitively the reasons are not obvious, and as it relies on some rather delicate calculations (which we know are sensitive to choices of coefficients in filters, etc). At the moment I

cannot be confident that the results are 'stable' enough to justify the conclusion that "it is important to distinguish near and far-field sources in the filtering approach".

If the authors can provide some 'conceptual backing' to support these results, then in my judgement the paper should clearly be accepted for publication in NHESS.

In saying this, please note that I accept the fact that some aspects of these filtering approaches cannot be completely stable (e.g. in the authors example, results with $H_{max} < 1m$ are not meaningful). This is expected, and not a problem. However, they need to provide more justification that the results at higher return periods are stable enough to justify the key conclusion.

**# Specific comments**

- P7, around line 10: I think you mean that you neglect a bunch of 'other' important sources of uncertainty, but, you do comprehensively test the filtering procedure (??right?? – actually upon reading the full paper I'm still uncertain). At the moment the paragraph doesn't make it clear if your example is actually a 'strong' evaluation of the filtering procedure, given the idealized assumptions on the source. Please make this clearer.

P8, top of page – it would be good to report on some sensitivity analysis of this to give the reader a 'feel' for how severe these approximations are (e.g. you could halve the number of clusters, so you don't have to do more simulations).

P8, bottom of page – 'it is worth noting that results at $H_{max} < 1m$ ...' – OK, but because those results are not meaningful, can you please 'clip' your figure limits so that they do not include $H_{max} < 1m$. That will help the reader focus on parts of the curve that you do consider meaningful, and ease the interpretation of the figures.

P9, paragraph around lines 10-15 – It's not evident to me why method 3a should 'over-estimate' rates for $H_{max}>3$ (or indeed why the difference is reversed at lower $H_{max}$). Can you give a heuristic explanation of why this could happen? Without

some idea of this, my thinking is 'maybe a calculation/convergence type error' (!). Or is it that, for large enough $H_{max}$, the associated local sources have a greater tendency to be filtered than the distant ones, for some reason – and the converse for smaller $H_{max}$? Definitely not obvious to me – please discuss it.

P9, paragraph around line 5 – I agree that you've shown that a 'blind' cluster analysis might produce quite different results from the 2-stage approach proposed in the paper. However, I'm less confident about the stability of either procedure. Can you really say that the 2-stage approach is better, based on the results presented here? Consider the following "devil's advocate" theory – from what you've presented, I hypothesis that "Both of your approaches are strongly affected by the details of the filtering coefficients, and equally big differences could be expected from merely adjusting those in reasonable ranges". In other words, how can readers be confident that the results are not just 'noise'? Probably you can justify this, but I don't see it from the current text. So please add in some discussion that explains 'why' these results happen, and why you expect them to be 'basically robust' {notwithstanding that you have to make some severe approximations for low events – that's ok – but at least for high events, we need some conceptual explanation of the results}.

P11, line 6 – as mentioned above, please provide more 'conceptual explanation' as to why this happens.

**# Detailed comments**

- P3, L31 – suggest changing 'is released' to 'is not used'.

- P5, L5: – suggest changing 'will produce as well similar inundation patterns' to 'will also produce similar inundation patterns'.

- P5, lines 6-7 – Please provide the equation for the cost function. I looked up the 2010 paper, but it appears to refer to time-series comparisons rather than H-max comparisons. Better to make it very obvious to the reader.

- P6, around lines 10-11 – It's not clear to me how you use the co-seismic deformation as a metric for source-proximity in the cluster analysis. Ahh, I see you do this below around lines 25. Give that, please add "(details below)" at the end of the sentence that finishes on line 11.

P7, line 20 – there is a number with multiple '.' inside – this is not familiar notation to me, do you intend to use some other separator?
* * *

---

## Referee Comment (RC2) · Anonymous Referee #2 · 7 Aug 2018

**##General comments**

This paper addresses an important topic, namely the development of onshore probabilistic tsunami hazard assessments and overcoming the related computational challenges. It builds on the work of Lorito et al. 2015 and Selva et al. 2016. A key innovation in this study is efficient filtering of near-field sources based on coseismic deformation, rather than offshore tsunami wave height. Overall, the paper is well written and concisely explains the issues and methods used to overcome them, and is suitable for publication in NHESS with some minor revisions.

In reviewing the paper, my main suggestions (details given below) are:

1. Siting the introduction more broadly in the PTHA literature. While this paper builds directly on the work of Lorito et al. 2015 and Selva et al. 2016, which is heavily relied upon in the introduction, along with the review paper by Grezio et al 2017, there are a number of additional relevant papers related to PTHA problems that could be cited. In my opinion, this would more neatly place this paper within the broader context of PTHA literature, widening the appeal of the paper. I.e. this paper should be framed as a step forward in PTHA in general, not just an update of the Lorito and Selva methods (although it is that too).

2. Some assessment of the sensitivity to the choices made in the filtering process (i.e. choice of thresholds etc) and whether this has any implication to the broader conclusions. Also whether it is possible for biases to be introduced in this process.

3. Some comment on whether other metrics besides maximum tsunami height or coseismic deformation could be relevant in assigning events to clusters.

In addition, there are several minor areas for clarification to improve the communication of the results, and a few grammatical errors.

**Specific comments**

1. Introduction

As mentioned above, this could benefit from reference to broader PTHA literature, specifically:

P2L4: Should also cite other PTHA studies as incremental gains in uncertainty quantification have been made over the past decade or so. Include Burbidge et al 2008; Gonzalez et al 2009; Horspool et al 2014; Davies et al 2017, Power et al 2017 (there may be others).

P2L7-8: These references (Geist and Lynett 2014; Grezio et al 2017) are not the first to emphasise computational approaches to PTHA – see additional references suggested in the above point.

NHESSD
P2L10: Should also reference Davies et al. 2017 regarding uncertainty quantification.

P2L13: Gonzalez et al 2009 should be cited in reference to challenges of PTHA for inundation.

P2L16: Geist 2002 should also be mentioned here.

P2L17. Mueller et al 2014 and Griffin et al. 2015 have both undertaken onshore tsunami hazard assessments considering heterogeneous earthquake rupture; although neither was fully probabilistic, they should be mentioned here as first steps towards quantifying this uncertainty for inundation hazard. Both also discuss the effect of coesismic displacement on onshore hazard and how this can vary locally, as discussed on P3L2. Here (P3L2) the discussion could be expanded to provide greater justification to your methodological approach to near field hazard.

Other comments on the introduction

P1L20: This isn't true. In practice many inundation assessments also use 'representative scenarios' for a range of return periods, not just 'worst credible'.

P1L22: One or a limited range of inundation scenarios get used for much more than 'a first screening' by emergency managers. These scenarios regularly get used to develop emergency management plans, evacuation plans, undertake impact assessments and so on. In my opinion this paragraph severely underplays the utility of scenario hazard assessments. The main problem is that we can't translate the offshore probability to an onshore probability. I expect that even with probabilistic inundation hazard maps, single event scenarios will still be used for a range of emergency management scenario planning purposes – we'll just be in a position to actually say what the probability of the event in terms of inundation hazard is.

P2L20: Need to clarify that this is talking about onshore PTHA – offshore PTHA are in general computationally affordable (though not cheap!) these days.

P2L30: 'while solving all the emerging technical and scientific issues'. This seems a
fairly bold claim! Perhaps rephrase.

- 2. Method outline This section is clear and well-written
- 3. Implementation of an improved filtering methodology

P5L4-5: How confident are you in the assumption that similar wave heights lead to similar onshore hazard? What about other wave properties such as period, which may be significant in determining onshore behaviour. E.g. Satake et al 2013 showed how inundation from the Tohoku tsunami was variably controlled by long-period components on flat coastal plains and shorter-period peaks in steep coastal areas. While set within a tsunami warning context rather than hazard assessment context, Gusman et al 2014 used two cycles of a tsunami waveform in identifying 'similar' tsunami. I think some of the issues are resolved for near field tsunami in your coseismic deformation filtering approach presented following, but it could still be good to comment on this issue here.

P6L30-35: It is not entirely clear how the distance is measured across the grid of coseismic deformation points, and how the spatial component is handled – perhaps also write the relevant equation to ensure clarity.

4. The Milazzo oil refinery

P8L28: The abbreviation Mmax is very commonly used to mean the maximum magnitude for a given earthquake source in seismic and tsunami hazard assessment. I would suggest changing this to something else to avoid confusion.

P9L11: This should be 'overestimates the probability for a given Hmax relative to STEP (3b).

P9L23: Should these be >=, not =, if you're talking about probabilities of exceedance?

P9L26-30: Use of phrase 'postitive' and 'negative differences' is confusing and makes the meaning of the paragraph somewhat ambiguous. Better to rephrase stating more explicitly which model gives relatively higher/lower hazard etc. Also, the difference be-
tween results far inland, near the coast and offshore in Figure 4a need to be explained. Why the shift from negative to postitive differences at some distance inland from the coast?

P10L5: Can anything additional be said about possible biases in the sampling process? Why is it likely that the sampling produced a non-representative selection of the important scenarios? How does this overall affect the strength of you conclusions in comparing the two methods (i.e. could the differences be random rather than systematic).

5. Conclusions P10L10: The statement around the definition of the source scenarios seems a bit strong. I'd suggest removing the word 'fully' as I doubt this has really been done. Aleatory uncertainty applies to both the rate model and the source location, geometry, maximum magnitude etc. I'd suggest putting 'and their mean annual rates' prior to 'exploring source uncertainty'.

P10L19: Suggest 'from offshore wave amplitudes alone'. Also, what about other parameters such as period for non near-field tsunami? This links back to my comments on Section 3.

Figures:

Figure 1: Step 2 should read 'tsunami propagation to offshore points'

Figures 3-5 need labels for parts a), b) etc.

**Technical corrections**

Throughout: Why use STEP instead of Step?

P1L11: demonstrate not demonstrated

P2L25: Rephrase to 'This allows identification of a subset of...'

P2L29: Rephrase to 'Here we merge the two approaches of Lorito et al. ...'
P3L21: Change 'resume' to 'summarise'. Also P10L9

P4L8: Change 'enough representative' to representative enough'

P6L10: Change 'and a separate modelling' to 'and separate modelling'

P6L34-35: Change to 'while the stopping criterion is based on the Euclidian distance'

P7L16: Mediterranean Sea (not sea)

P8L1: Replace 'Namely' with 'That is'; delete 'even'

P8L25: Please specify the shear modulus used for the Okada calculations

P9L15: Remove 'supposedly'

P10L24: Change 'has not to be' to 'is not'.

P11L3-4: I think this should read 'As a consequence, the effect of coastal deformation on tsunami hazard can not be deduced...'

P11L14: Change to '...the approach developed here allows consideration of a very high number...'

References

References mentioned above not already in the manuscript:

Burbidge, D., Cummins, P.R., Mleczko, R. and Thio, H.K., 2008. A probabilistic tsunami hazard assessment for Western Australia. In Tsunami Science Four Years after the 2004 Indian Ocean Tsunami (pp. 2059-2088). Birkhäuser Basel.

Davies, G., Griffin, J., Løvholt, F., Glimsdal, S., Harbitz, C., Thio, H.K., Lorito, S., Basili, R., Selva, J., Geist, E. and Baptista, M.A., 2017. A global probabilistic tsunami hazard assessment from earthquake sources. Geological Society, London, Special Publications, 456, pp.SP456-5.

Geist, E.L., 2002. Complex earthquake rupture and local tsunamis. Journal of Geo-
physical Research: Solid Earth, 107(B5).

Griffin, J.D., Pranantyo, I.R., Kongko, W., Haunan, A., Robiana, R., Miller, V., Davies, G., Horspool, N., Maemunah, I., Widjaja, W.B. and Natawidjaja, D.H., 2016. Assessing tsunami hazard using heterogeneous slip models in the Mentawai Islands, Indonesia. Geological Society, London, Special Publications, 441, pp.SP441-3.

Gusman, A.R., Tanioka, Y., MacInnes, B.T. and Tsushima, H., 2014. A methodology for near-field tsunami inundation forecasting: Application to the 2011 Tohoku tsunami. Journal of Geophysical Research: Solid Earth, 119(11), pp.8186-8206.

Horspool, N., Pranantyo, I., Griffin, J., Latief, H., Natawidjaja, D.H., Kongko, W., Cipta, A., Bustaman, B., Anugrah, S.D. and Thio, H.K., 2014. A probabilistic tsunami hazard assessment for Indonesia. Natural Hazards and Earth System Sciences, 14(11), pp.3105-3122.

Power, W., Wang, X., Wallace, L., Clark, K. and Mueller, C., 2017. The New Zealand Probabilistic Tsunami Hazard Model: development and implementation of a methodology for estimating tsunami hazard nationwide. Geological Society, London, Special Publications, 456, pp.SP456-6.

Satake, K., Fujii, Y., Harada, T. and Namegaya, Y., 2013. Time and space distribution of coseismic slip of the 2011 Tohoku earthquake as inferred from tsunami waveform data. Bulletin of the seismological society of America, 103(2B), pp.1473-1492.

NHESSD

---

## Referee Comment (RC3) · Anonymous Referee #3 · 15 Aug 2018

Review for NHESS-2018-202 entitled: "From regional to local SPTHA: efficient computation of probabilistic inundation maps addressing near-field sources" by Manuela Volpe et al.

Overview: The authors did an extension of the SPTHA method previously proposed by Lorito et al. (2015) [GJI] and Selva et al. (2016) [GJI]. A new filtering scheme for earthquake scenarios is developed (Filter P) and the method is applied to a particular coast (Millazo).

Overall evaluation: The application of the SPTHA to a new region and adding some innovations to the previously-developed method may justify publication of this work.

The paper is not as good as the other two papers published before (I mean Lorito et al. 2015; Selva et al. 2016). The current manuscript refers to the previous two papers very frequently and does not seem to stand by its own. However, I am positive about this work and I think it can be published in NHESS after some revisions. I made some suggestions below.

Comments: - Page 3, "Method": your Section 2 looks a review of the methods previously published by Lorito et al. (2015) [GJI] and Selva et al. (2016) [GJI]. Your own method is outlined in Section 3. This is confusing. In fact, your current section 2 is sort of literature review. I suggest change the title of Section 2 to "A review of SPTHA" and then change title of Section 3 to "Methods: an improved SPTHA".

- To show the better performance of the new method over the ones published before (I mean Lorito et al. 2015; Selva et al. 2016), a discussion or a figure is needed.

- Try not to refer to two previous papers so much. You may want to show more independence.

- In Page 9, refer to appropriate figures when discussing the results.

- Why you have capital letters for STEP? Is that necessary? If not, change it to "step" because when you use capital letters, the reader assumes it is an acronym. I guess it is not an acronym for anything.

- Page 5, Line 17: explain more about Filter P.

- Page 6, Line 2: what is intra-cluster? It is unclear. Make sure to explain more about it and clarify how it works.

Page 6, Line 7: delete statements like "as mentioned before..." it is not suitable for academic writing.

Page 7, Line 32: delete "as discussed in previous ...." Again not suitable.

Page 8: here you use "cluster" and "scenario" interchangeably. Make sure which one

you meant. I assume that you meant "Scenario" not "cluster". They are different. Cluster is much bigger than a single scenario. One cluster can include 200 scenarios. In Line 13, you say: "We obtained 634 and 520 clusters for remote and local sources, respectively, that is a total of 1154 scenarios. . .". Here the sum of 634 and 520 clusters cannot be 1154 scenarios. Instead, the sum of 634 and 520 clusters cannot be 1154 CLUSTERS.

- Your conclusion has many repeats; for example lines 8-13. Make sure delete all repeats.

- ABSTRACT: try to have more numbers and conclusions, not only generic statements.

- CONCLUSIONS: shorten it to a paragraph and be specific and do not repeat all stuff again.

- Figure 1: The last box repeats. Delete one of them.

- Figure 2: Explain what are two sets of red dots.

End of review.

---

## Author Comment (AC1) · 25 Oct 2018

Dear Reviewer,

we thank you for your thoughtful comments. We addressed all of them as specified in detail in the point-to-point answers in the supplement pdf file, both in response to the general and to the specific comments of yours.

Here, we make some general remarks, since we made one important change in the revised manuscript. We also ask for a minor change to the title to insert in it the word "tsunami", which was missing in the original title. This letter is repeated in all the three

responses to the three Reviewers.

Changes to the results and to the manuscript.

We first of all need to point out that the change we made was necessary, since we found a bug in one of the numerical codes we had written for this study. This bug was found while performing some tests, some of them conceived for addressing your comments, particularly as far as the robustness and the importance of the correction for the near-field sources compared to the "noise" introduced by the tuning of the various filtering thresholds were concerned.

The bug consisted in a missing sum operator in the computation of the cluster probability (a missing cycle over one variable!). Hence, the probability of the entire cluster was not assigned to the cluster representative.

The new results, computed after the bug was corrected, do not differ in essence, although the resulting probabilities are obviously overall higher. All the new Figures are enclosed.

Conversely, for hazard intensities higher than 1 meter, the results now show even more pronounced differences between the "corrected" and "uncorrected" filtering procedures (new Figure 3c).

Our results now more clearly point out that not considering an appropriate correction for the near field would lead to overestimate the tsunami hazard. This is true in the case of this specific setting, though, since we found a prevalence of clusters causing coastal uplift from the near-field sources (the situation may be the opposite as well or a mix for different source-coast configuration).

These uplifts would tend to diminish tsunami inundation. Hence, the tsunami hazard would be overestimated without taking this into account. We hope this is illustrated by the new Figure 3c.

We now in fact added to panel c of this Figure a new quantity, that is the Mean Uplift

(hereinafter MU) on a random point on the coastline in the inner - highest resolution - grid domain.

The MU provides the mean - over all scenarios contributing to some hazard intensities and all coastal points - co-seismic coastal displacement (with positive sign if uplift) plotted versus different hazard intensities, and it is compared to the relative percentage differences between the corrected and uncorrected results.

In more detail, the MU is obtained:

1. for the mean model - the same considered before as far as epistemic uncertainty is concerned;

2. for each single hazard intensity threshold, as said;

3. by performing a weighted average of the uplifts from each model (represented through the centroid of the cluster), where the weights are the annual probabilities of the individual models (of the individual earthquakes then), set to zero if the earthquake do not deform the coastline (i.e. for far-field sources) or if the tsunami doesn't exceed the given hazard intensity threshold; the weighted average is normalized to the total probability of the near- and far-field sources contributing to the tsunami hazard for that threshold;

4. by further averaging the result along the coastline, hence the MU may be interpreted as the mean on a random point of the coastline, from all the far-field and near-field scenarios, the latter including those causing both subsidence and uplift (note than that the absolute MU value in meters is then rather small being averaged over sources that cause either uplift or subsidence, or no coastal displacement at all).

Note that the intermediate results (before applying item 4.), that is the MU on each coastal point, for different intensity thresholds, both for single cluster representatives (red lines) and for the weighted average (according to item 3., blue line) are plotted in the new Figure S4. While we note that there are both positive and negative displacements (red lines corresponding to uplift and subsidence along the coast respectively), the predominant one is unveiled by the sum over the different clusters plotted (the a blue line).

Moreover, the results in Figure 3c now show very little differences between the "corrected" and "uncorrected" filtering procedures at low hazard intensities, that is those below the Filter H thresholds value of 1 meter.

In summary, for the specific case study, that is for this specific source-target configuration, our findings show that not considering an appropriate correction for near field would lead to overestimate the tsunami hazard for Hmax greater than 1m, and this overestimation is correlated to dominant coastal uplift. At lower intensities differences are small but not meaningful, as the results are biased by Filter H.

We will of course add the necessary new text in the revised manuscript concerning MU and the corresponding analysis.

New title.

We propose the following new title for this study: "From regional to local SPTHA: efficient computation of probabilistic tsunami inundation maps addressing near-field sources" That is, we just inserted the word "tsunami" before inundation. This would make easier to find the article if searching NHESS for tsunami-related papers.

Kind Regards, Manuela Volpe, on the behalf of the co-authors.

Please also note the supplement to this comment:
https://www.nat-hazards-earth-syst-sci-discuss.net/nhess-2018-202/nhess-2018-202-AC1-supplement.pdf

[Figure]

[Figure]

**Fig. 1.** Figure 3

[Figure]

**Fig. 2.** Figure 4

[Figure]

**Fig. 3.** Figure 5

**Fig. 4.** Figure S2

[Figure]

[Figure]

[Figure]

[Figure]

**Fig. 5.** Figure S3

**Fig. 6.** Figure S4

**Supplement:**

**Response point-by-point to Anonymous Referee #1**

The point-by-point answers are in blue color, below each Reviewer's comment (reported in *Italic*).

*# # Overall comments #*

*This looks to be a good paper about an important topic that is of clear interest to NHESS readers. It is mostly well written, and describes innovative ideas which are likely to be of broad utility in tsunami hazard assessment.*

> *My only 'significant' concern is that the authors do not provide a 'conceptual justification' for the differences in the results of the two filtering methods they apply, for the case of high $H_{max}$. As it stands, as a reader I don't know why this happens. Intuitively the reasons are not obvious, and as it relies on some rather delicate calculations (which we know are sensitive to choices of coefficients in filters, etc). At the moment I cannot be confident that the results are 'stable' enough to justify the conclusion that "it is important to distinguish near and far-field sources in the filtering approach". If the authors can provide some 'conceptual backing' to support these results, then in my judgement the paper should clearly be accepted for publication in NHESS. In saying this, please note that I accept the fact that some aspects of these filtering approaches cannot be completely stable (e.g. in the authors example, results with $H_{max} < 1m$ are not meaningful). This is expected, and not a problem. However, they need to provide more justification that the results at higher return periods are stable enough to justify the key conclusion.*

(the answer below is the same for a similar question from Reviewer 2)

The conceptual explanation traces back to the fact that the two procedures are not equivalent from a physical point of view and we could roughly say that one is in principle "correct" and the other one is "wrong". Maybe in saying "it is important to distinguish near and far-field sources in the filtering approach" we were not clear enough. What we wanted to stress is that a blind filtering procedure based on offshore tsunami amplitudes produces a non representative selection of the important scenarios, as it could aggregate or even remove important local scenarios.

We try to explain it better below.

In the original procedure by Lorito et al., offshore tsunami amplitudes are supposed to be representative of the coastal inundation, regardless of the source location with respect to the coast. That was reasonable, since it considered either far field scenarios with respect to the coast of Sicily, or scenarios which deformed the coast of Crete Island always in the same direction, since they were all subduction earthquake on the neary Hellenic Arc.

Indeed, offshore tsunami profiles could be strongly misleading when coseismic deformation of the coast occurs, either as coastal uplift or subsidence depending on the causative earthquake. The coseismic displacement induced by local earthquakes can modify the actual onshore tsunami intensity corresponding to the same offshore wave. Hence, near field scenarios must be separately treated, and clustered considering the source similarities, including the co-seismic coastal displacement, rather than the offshore tsunami wave similarity.

We will try to report these "conceptual" arguments in the revised manuscript as concisely as possible.

The tuning of the thresholds in the filtering procedure is a different task, but we note that the same thresholds have been used with and without the correction for near field, so that the differences we found in the results obtained from the two procedures are not in our opinion imputable to those choices.

On the other hand, we can now support such conceptual justification providing the physical explanation of the specific results, based on the new quantity MU (mean uplift) we calculated and described in our introductive general remarks. This also answers to one of your specific comments below.

In general, lower 'corrected' hazard means that the predominant effect by local sources contributing to a specific point on the hazard curve - that is to the probability of exceedance for a given intensity threshold - is represented by coastal uplift, which in turn decreases tsunami hazard. In other words, there is a prevalence of clusters represented by scenarios causing uplift. Conversely, higher hazard would correspond to coastal subsidence.

As we said, we investigated this aspect, computing, for different intensity thresholds above 1m, the MU on a random point along the coastline of the inner grid, produced by near field representative scenarios contributing to the hazard at that threshold, weighted by the occurrence probability associated to each scenario (corresponding to the probability of the entire cluster it represents) and normalized to the probability of all of the scenarios contributing to the same intensity threshold.

The obtained positive values, although not representative of the real coastal displacement as averaged on all the scenarios (including that ones which do not produce appreciable coseismic local deformation), indicate that the dominant contribution to the coseismic deformation is an uplift of the coast, in agreement with the percentage differences retrieved between the two approaches.

We hope to have answered in this way to the "significant" concern expressed by the Reviewer. We must acknowledge that this comment made us deepen the analysis and consider our results much more carefully - and indeed we found a bug.

> ➢ *P7, around line 10: I think you mean that you neglect a bunch of 'other' important sources of uncertainty, but, you do comprehensively test the filtering procedure (??right?? – actually upon reading the full paper I'm still uncertain). At the moment the paragraph doesn't make it clear if your example is actually a 'strong' evaluation of the filtering procedure, given the idealized assumptions on the source. Please make this clearer.*

The test site illustrative application is not a real hazard assessment, as it is based on a quite rough probability model as well as on some strong assumptions regarding the filtering thresholds and the source modeling.

Nevertheless, although relatively simplified, our source model is still quite complex, and includes even epistemic uncertainties on many source parameters, e.g concerning the seismic rates, the shape of the magnitude-frequency distribution, even the seismogenic depth for the two considered subduction zones, and several others. It also includes ensemble uncertainty modeling. We now include a new Figure in the Supplementary Materials (Figure S2), which should make clearer that the model deals with epistemic uncertainty, as it shows the comparison between the mean offshore hazard curves at selected points along the 50m isobath (see Figure 2a of the manuscript), as well as the comparison between some quantiles of the epistemic uncertainty, for the filtered and original set of scenarios. Please refer to Selva et al. 2016 for further details on the adopted source model.

So, we consider the model as fully suitable to test and describe the procedure. We anyway restate that the aim of the application is to highlight that inaccurate (biased) evaluation of site-specific tsunami hazard would be obtained if scenarios located in the near field of the target area are not properly taken into account, irrespectively of the completeness and consequent complexity of the hazard assessment. A "real" application would be just more complicated and more computationally demanding.

> ➢ P8, top of page – it would be good to report on some sensitivity analysis of this to give the reader a 'feel' for how severe these approximations are (e.g. you could halve the number of clusters, so you don't have to do more simulations).

The filtering procedure surely introduces some approximations and ideally the goal should be to reduce the computational cost of PTHA while keeping the error with respect to the whole set of sources as small as possible. In the present work, considering the illustrative nature of the case study, we enlarged the accepted error to further reduce the number of explicit numerical simulations.

First of all, the most severe approximation was made during the filtering on tsunami amplitude: it goes without saying that a threshold of 1m might be not acceptable in case of a real hazard assessment, while it is an acceptable threshold for illustrative purposes. It is worth stressing that this filter, independently from the threshold value, does not affect

subsequent steps of the procedure, as it represents a rigid cut-off of the number of scenarios we are accounting for.

Another strong assumption was made regarding the cluster analysis: the k-medoids partitioning algorithm is based on the minimization of the sum of the intra-cluster distances, i.e. the distances between each element of a cluster and the cluster centroid. Strong constraints on the distances result in a more accurate partitioning, in terms of similarity between the elements of each cluster, but lead to a great number of clusters. Instead, larger ranges of acceptability increase the efficiency of the algorithm, in terms of number of resulting clusters, to the detriment of the accuracy.

As an example, we provide here a sensitivity analysis on the threshold imposed on the intra-cluster variance (step (3a)): the new Figure S3 shows the relative differences in absolute value between the offshore (i.e. at the control points along the 50m isobath) hazard curves computed from the complete initial set of sources and the filtered set (at the end of the cluster analysis). The red box corresponds to the threshold value we chose (0.2): it appears evident that a smaller value would have allowed a stronger constraint on the error introduced by the cluster analysis, while considerably increasing the number of resulting clusters. Vice versa, higher thresholds produce a smaller number of clusters, but fail in reproducing the hazard (error up to 40%). Our choice in our opinion represented the best trade off for our purposes. Again, in case of a real hazard assessment, the lower threshold would be likely better.

> *P8, bottom of page – 'it is worth noting that results at $H_{max} < 1m$ ...' – OK, but because those results are not meaningful, can you please 'clip' your figure limits so that they do not include $H_{max} < 1m$. That will help the reader focus on parts of the curve that you do consider meaningful, and ease the interpretation of the figures.*

We apologise as we have misspoken: what we intended is that the hazard curves below 1m can be (negatively) biased since they are depleted from the scenarios removed by Filter H. Following your suggestion, we rephrased and shadowed that part of the plots in Figure 3. This depletion is also clearly observed in new Figure S2 for low amplitudes.

> *P9, paragraph around lines 10-15 – It's not evident to me why method 3a should 'over-estimate' rates for $H_{max} > 3$ (or indeed why the difference is reversed at lower $H_{max}$). Can you give a heuristic explanation of why this could happen? Without some idea of this, my thinking is 'maybe a calculation/convergence type error' (!). Or is it that, for large enough $H_{max}$, the associated local sources have a greater tendency to be filtered than the distant ones, for some reason – and the converse for smaller $H_{max}$? Definitely not obvious to me – please discuss it.*

As discussed before, a lower hazard at a certain point of the hazard curve, due to the near field correction, means that the coseismic field from local sources that dominate the hazard produces a coastal uplift. We agree this was not sufficiently proved before, but the new Figure (3c), with the MU superimposed to the percentage differences between the two

approaches now should better illustrate that the overestimation is correlated to the dominant coastal uplift.

> *P9, paragraph around line 5 – I agree that you've shown that a 'blind' cluster analysis might produce quite different results from the 2-stage approach proposed in the paper. However, I'm less confident about the stability of either procedure. Can you really say that the 2-stage approach is better, based on the results presented here? Consider the following "devil's advocate" theory – from what you've presented, I hypothesis that "Both of your approaches are strongly affected by the details of the filtering coefficients, and equally big differences could be expected from merely adjusting those in reasonable ranges". In other words, how can readers be confident that the results are not just 'noise'? Probably you can justify this, but I don't see it from the current text. So please add in some discussion that explains 'why' these results happen, and why you expect them to be 'basically robust' {notwithstanding that you have to make some severe approximations for low events – that's ok – but at least for high events, we need some conceptual explanation of the results}.*

As it should be clear now from our previous answers, involving the new Figure 3 and MU, there are firm conceptual reasons supporting the need of a "2-stage approach". You are indeed right that the results are stronger for larger amplitudes. This is clearer with the new results.

The simplest explanation remains though the same: the original assumption that offshore tsunami amplitudes are representative of the coastal inundation may fail if local sources producing appreciable coseismic deformation of the coast - of conflicting sign, i.e. some uplift and some subsidence depending on the source - are involved.

Hence, one (the only?) way we can take this into account is to separate near and far field scenarios and treat local sources removing the approximation introduced by the cluster analysis on the tsunami amplitudes, since offshore profile can not be considered reliable for such sources. This considerations hold irrespectively of the stability of the results in our example. However, our results "behave" as expected, being dependent on the approach used.

Indeed, the near field treatment is still an approximation, as we reduced the number of numerical simulations with respect to the "exact" case, by performing a cluster analysis based on the coseismic fields. However, it should be a better approximation with respect to aggregate local and remote scenarios on the basis of the offshore tsunami amplitudes.

Moreover, our application is also aimed to investigate if such procedure is really needed from the point of view of results, that is if, apart from the physical meaning of the procedure, results are actually affected by the near field correction.

The fact that the contribution from the near field turned out to be significant, even investigating a target site with relatively low near-field tsunamigenic seismicity, was not straightforward.

We finally stress that the two approaches (with or without the correction for near field) only differ in the way local sources are treated: the filtering coefficients are basically consistent. In other words, the different results can not be related to the filtering thresholds.

We repeat, the "conceptual explanation", should now be there with the new results and the new analysis presented. And we must acknowledge that this comment was really useful.

> ➢ P11, line 6 – as mentioned above, please provide more 'conceptual explanation' as to why this happens.

As extensively discussed in the previous answers, this is related to the coseismic deformation induced by local sources, which, if properly accounted, modify the effective tsunami hazard.

**# Detailed comments**
> ➢ P3, L31 – suggest changing 'is released' to 'is not used'.
> ➢ P5, L5: – suggest changing 'will produce as well similar inundation patterns' to 'will also produce similar inundation patterns'.
> ➢ P5, lines 6-7 – Please provide the equation for the cost function. I looked up the 2010 paper, but it appears to refer to time-series comparisons rather than H-max comparisons. Better to make it very obvious to the reader.
> ➢ P6, around lines 10-11 – It's not clear to me how you use the co-seismic deformation as a metric for source-proximity in the cluster analysis. Ahh, I see you do this below around lines 25. Give that, please add "(details below)" at the end of the sentence that finishes on line 11.
> ➢ P7, line 20 – there is a number with multiple '.' inside – this is not familiar notation to me, do you intend to use some other separator?

We thank for your suggestions, which will be all addressed in the revised manuscript. In particular, we will clarify that the cost function equation firstly introduced in the 2010 papers to compare time series while solving inverse problems was modified by Lorito et. al 2015 for Hmax comparison.

---

## Author Comment (AC2) · 25 Oct 2018

Dear Reviewer,

we thank you for your thoughtful comments. We addressed all of them as specified in detail in the point-to-point answers in the supplement pdf file, both in response to the general and to the specific comments of yours.

Here, we make some general remarks, since we made one important change in the revised manuscript. We also ask for a minor change to the title to insert in it the word "tsunami", which was missing in the original title. This letter is repeated in all the three

responses to the three Reviewers.

Changes to the results and to the manuscript.

We first of all need to point out that the change we made was necessary, since we found a bug in one of the numerical codes we had written for this study. This bug was found while performing some tests, some of them conceived for addressing your comments, particularly as far as the robustness and the importance of the correction for the near-field sources compared to the “noise” introduced by the tuning of the various filtering thresholds were concerned.

The bug consisted in a missing sum operator in the computation of the cluster probability (a missing cycle over one variable!). Hence, the probability of the entire cluster was not assigned to the cluster representative.

The new results, computed after the bug was corrected, do not differ in essence, although the resulting probabilities are obviously overall higher. All the new Figures are enclosed.

Conversely, for hazard intensities higher than 1 meter, the results now show even more pronounced differences between the “corrected” and “uncorrected” filtering procedures (new Figure 3c).

Our results now more clearly point out that not considering an appropriate correction for the near field would lead to overestimate the tsunami hazard. This is true in the case of this specific setting, though, since we found a prevalence of clusters causing coastal uplift from the near-field sources (the situation may be the opposite as well or a mix for different source-coast configuration).

These uplifts would tend to diminish tsunami inundation. Hence, the tsunami hazard would be overestimated without taking this into account. We hope this is illustrated by the new Figure 3c.

We now in fact added to panel c of this Figure a new quantity, that is the Mean Uplift

(hereinafter MU) on a random point on the coastline in the inner - highest resolution - grid domain.

The MU provides the mean - over all scenarios contributing to some hazard intensities and all coastal points - co-seismic coastal displacement (with positive sign if uplift) plotted versus different hazard intensities, and it is compared to the relative percentage differences between the corrected and uncorrected results.

In more detail, the MU is obtained:

1.for the mean model - the same considered before as far as epistemic uncertainty is concerned;

2. for each single hazard intensity threshold, as said;

3. by performing a weighted average of the uplifts from each model (represented through the centroid of the cluster), where the weights are the annual probabilities of the individual models (of the individual earthquakes then), set to zero if the earthquake do not deform the coastline (i.e. for far-field sources) or if the tsunami doesn't exceed the given hazard intensity threshold; the weighted average is normalized to the total probability of the near- and far-field sources contributing to the tsunami hazard for that threshold;

4. by further averaging the result along the coastline, hence the MU may be interpreted as the mean on a random point of the coastline, from all the far-field and near-field scenarios, the latter including those causing both subsidence and uplift (note than that the absolute MU value in meters is then rather small being averaged over sources that cause either uplift or subsidence, or no coastal displacement at all).

Note that the intermediate results (before applying item 4.), that is the MU on each coastal point, for different intensity thresholds, both for single cluster representatives (red lines) and for the weighted average (according to item 3., blue line) are plotted in the new Figure S4. While we note that there are both positive and negative displacements (red lines corresponding to uplift and subsidence along the coast respectively), the predominant one is unveiled by the sum over the different clusters plotted (the a blue line).

Moreover, the results in Figure 3c now show very little differences between the "corrected" and "uncorrected" filtering procedures at low hazard intensities, that is those below the Filter H thresholds value of 1 meter.

In summary, for the specific case study, that is for this specific source-target configuration, our findings show that not considering an appropriate correction for near field would lead to overestimate the tsunami hazard for Hmax greater than 1m, and this overestimation is correlated to dominant coastal uplift. At lower intensities differences are small but not meaningful, as the results are biased by Filter H.

We will of course add the necessary new text in the revised manuscript concerning MU and the corresponding analysis.

New title.

We propose the following new title for this study: "From regional to local SPTHA: efficient computation of probabilistic tsunami inundation maps addressing near-field sources" That is, we just inserted the word "tsunami" before inundation. This would make easier to find the article if searching NHESS for tsunami-related papers.

Kind Regards, Manuela Volpe, on the behalf of the co-authors.

Please also note the supplement to this comment:
https://www.nat-hazards-earth-syst-sci-discuss.net/nhess-2018-202/nhess-2018-202-AC2-supplement.pdf
* * *
[Figure]

**Fig. 1.** Figure 3

[Figure]

**Fig. 2.** Figure 4

[Figure]

**Fig. 3.** Figure 5

[Figure]

**Fig. 4.** Figure S2

[Figure]

**Fig. 5.** Figure S3

[Figure]

**Fig. 6.** Figure S4

**Supplement:**

**Response point-by-point to Anonymous Referee #2**

The point-by-point answers are in blue color, below each Reviewer's comment (reported in *Italic*).

**General comments**

*This paper addresses an important topic, namely the development of onshore probabilistic tsunami hazard assessments and overcoming the related computational challenges. It builds on the work of Lorito et al. 2015 and Selva et al. 2016. A key innovation in this study is efficient filtering of near-field sources based on coseismic deformation, rather than offshore tsunami wave height. Overall, the paper is well written and concisely explains the issues and methods used to overcome them, and is suitable for publication in NHESS with some minor revisions.*

*In reviewing the paper, my main suggestions (details given below) are:*

1. *Siting the introduction more broadly in the PTHA literature. While this paper builds directly on the work of Lorito et al. 2015 and Selva et al. 2016, which is heavily relied upon in the introduction, along with the review paper by Grezio et al 2017, there are a number of additional relevant papers related to PTHA problems that could be cited. In my opinion, this would more neatly place this paper within the broader context of PTHA literature, widening the appeal of the paper. I.e. this paper should be framed as a step forward in PTHA in general, not just an update of the Lorito and Selva methods (although it is that too).*

We fully agree and appreciate the suggestion. Citing Lorito et al. 2015 and Selva et al. 2016 was indeed mandatory. Conversely, using only Grezio et al. 2017 to refer to PTHA was certainly too simplistic. We will improve the bibliography to better frame the paper in the context, also following your specific suggestions below.

As a result, we added the following references, including also those stemming from other comments below:

Brizuela, B., Armigliato, A., and Tinti, S. (2014). Assessment of tsunami hazards for the central american pacific coast from southern mexico to northern peru. Natural Hazards and Earth System Sciences, 14(7):1889–1903.

Burbidge, D., Cummins, P. R., Mleczko, R., and Thio, H. K. (2008). A probabilistic tsunami hazard assessment for western australia. Pure Appl. Geophys., 165(11):2059–2088.

Davies, G., Griffin, J., Løvholt, F., Glimsdal, S., Harbitz, C., Thio, H. K., Lorito, S., Basili, R., Selva, J., Geist, E., and Baptista, M. A. (2017). A global probabilistic tsunami hazard

assessment from earthquake sources. Geological Society, London, Special Publications, 456.

Gailler, A., Calais, E., Hebert, H., Roy, C., and Okal, E. (2015). Tsunami scenarios and hazard assessment along the northern coast of haiti. Geophysical Journal International, 203(3):2287–2302.

Geist, E. L. (2002). Complex earthquake rupture and local tsunamis. Journal of Geophysical Research: Solid Earth, 107(B5):ESE 2–1–ESE 2–15.

Griffin, J. D., Pranantyo, I. R., Kongko, W., Haunan, A., Robiana, R., Miller, V., Davies, G., Horspool, N., Maemunah, I., Widjaja, W. B., Natawidjaja, D. H., and Latief, H. (2017). Assessing tsunami hazard using heterogeneous slip models in the Mentawai Islands, Indonesia. Geological Society of London Special Publications, 441:47–70.

Gusman, A. R., Tanioka, Y., MacInnes, B. T., and Tsushima, H. (2014). A methodology for near-field tsunami inundation forecasting: Application to the 2011 tohoku tsunami. Journal of Geophysical Research: Solid Earth, 119(11):8186–8206.

Harbitz, C., Glimsdal, S., Bazin, S., Zamora, N., Løvholt, F., Bungum, H., Smebye, H., Gauer, P., and Kjekstad, O. (2012). Tsunami hazard in the caribbean: Regional exposure derived from credible worst case scenarios. Continental Shelf Research, 38:1 – 23.

Horspool, N., Pranantyo, I., Griffin, J., Latief, H., Natawidjaja, D. H., Kongko, W., Cipta, A., Bustaman, B., Anugrah, S. D., and Thio, H. K. (2014). A probabilistic tsunami hazard assessment for indonesia. Nat. Hazards Earth Syst. Sci., 14(11):3105–3122.

Løvholt, F., Bungum, H., Harbitz, C. B., Glimsdal, S., Lindholm, C. D., and Pedersen, G. (2006). Earthquake related tsunami hazard along the western coast of thailand. Natural Hazards and Earth System Sciences, 6(6):979–997.

Power, W., Wang, X., Wallace, L., Clark, K., and Mueller, C. (2017). The New Zealand probabilistic tsunami hazard model: development and implementation of a methodology for estimating tsunami hazard nationwide. Geological Society, London, Special Publications, 456.

Satake, K., Fujii, Y., Harada, T., and Namegaya, Y. (2013). Time and space distribution of coseismic slip of the 2011 tohoku earthquake as inferred from tsunami waveform datatime and space distribution of coseismic slip of the 2011 tohoku earthquake. Bulletin of the Seismological Society of America, 103(2B):1473.

2.  *Some assessment of the sensitivity to the choices made in the filtering process (i.e. choice of thresholds etc) and whether this has any implication to the broader conclusions. Also whether it is possible for biases to be introduced in this process.*

(the answer below is the same for a similar question from Reviewer 1)

The conceptual explanation traces back to the fact that the two procedures are not equivalent from a physical point of view and we could roughly say that one is in principle "correct" and the other one is "wrong". Maybe in saying "it is important to distinguish near and far-field sources in the filtering approach" we were not clear enough. What we wanted to stress is that a blind filtering procedure based on offshore tsunami amplitudes produces a non representative selection of the important scenarios, as it could aggregate or even remove important local scenarios.

We try to explain it better below.

In the original procedure by Lorito et al., offshore tsunami amplitudes are supposed to be representative of the coastal inundation, regardless of the source location with respect to the coast. That was reasonable, since it considered either far field scenarios with respect to the coast of Sicily, or scenarios which deformed the coast of Crete Island always in the same direction, since they were all subduction earthquake on the neary Hellenic Arc.

Indeed, offshore tsunami profiles could be strongly misleading when coseismic deformation of the coast occurs, either as coastal uplift or subsidence depending on the causative earthquake. The coseismic displacement induced by local earthquakes can modify the actual onshore tsunami intensity corresponding to the same offshore wave. Hence, near field scenarios must be separately treated, and clustered considering the source similarities, including the co-seismic coastal displacement, rather than the offshore tsunami wave similarity.

We will try to report these "conceptual" arguments in the revised manuscript as concisely as possible.

The tuning of the thresholds in the filtering procedure is a different task, but we note that the same thresholds have been used with and without the correction for near field, so that the differences we found in the results obtained from the two procedures are not in our opinion imputable to those choices.

On the other hand, we can now support such conceptual justification providing the physical explanation of the specific results, based on the new quantity MU (mean uplift) we calculated and described in our introductive general remarks. This also answers to one of your specific comments below.

In general, lower 'corrected' hazard means that the predominant effect by local sources contributing to a specific point on the hazard curve - that is to the probability of exceedance for a given intensity threshold - is represented by coastal uplift, which in turn decreases tsunami hazard. In other words, there is a prevalence of clusters represented by scenarios causing uplift. Conversely, higher hazard would correspond to coastal subsidence.

As we said, we investigated this aspect, computing, for different intensity thresholds above 1m, the MU on a random point along the coastline of the inner grid, produced by near field representative scenarios contributing to the hazard at that threshold, weighted by the occurrence probability associated to each scenario (corresponding to the probability of the entire cluster it represents) and normalized to the probability of all of the scenarios contributing to the same intensity threshold.

The obtained positive values, although not representative of the real coastal displacement as averaged on all the scenarios (including that ones which do not produce appreciable coseismic local deformation), indicate that the dominant contribution to the coseismic deformation is an uplift of the coast, in agreement with the percentage differences retrieved between the two approaches.

We hope to have answered in this way to the "significant" concern expressed by the Reviewer. We must acknowledge that this comment made us deepen the analysis and consider our results much more carefully - and indeed we found a bug.

>   3. *Some comment on whether other metrics besides maximum tsunami height or co-seismic deformation could be relevant in assigning events to clusters.*

Yes, sure. This might be certainly relevant, at least for far-field sources. Storing and using the full waveforms, or considering maybe periods and polarities, or other approaches, can be considered.

Take into account though that this was already briefly discussed in Lorito et al. 2015. It was tested there that after some tuning of the length of the offshore profile of control points, the offshore height profile turned out to be a sufficiently good indicator for approximating the inundation afterwards. We may speculate that this is due to the collective information provided by the maximum heights themselves taken altogether, which then becomes a kind of maximum wave profile. Nevertheless, we will briefly discuss the issue in the revised manuscript, also using the examples you provide in your specific comment.

Vice-versa, as far as near-field sources are concerned, two modelled tsunamis with very similar sources should be quite similar, except in case of a very sensitive dependence on initial conditions - like for the butterfly-effect. We are not totally convinced but we will cautiously mention the issue in the revised manuscript.

*In addition, there are several minor areas for clarification to improve the communication of the results, and a few grammatical errors.*

*##Specific comments##*

*1. Introduction*
*As mentioned above, this could benefit from reference to broader PTHA literature, specifically:*

> *P2L4: Should also cite other PTHA studies as incremental gains in uncertainty quantification have been made over the past decade or so. Include Burbidge et al 2008; Gonzalez et al 2009; Horspool et al 2014; Davies et al 2017, Power et al 2017 (there may be others).*

> *P2L7-8: These references (Geist and Lynett 2014; Grezio et al 2017) are not the first to emphasise computational approaches to PTHA – see additional references suggested in the above point.*

> *P2L10: Should also reference Davies et al. 2017 regarding uncertainty quantification.*

> *P2L13: Gonzalez et al 2009 should be cited in reference to challenges of PTHA for inundation.*

> *P2L16: Geist 2002 should also be mentioned here.*

> *P2L17. Mueller et al 2014 and Griffin et al. 2015 have both undertaken on-shore tsunami hazard assessments considering heterogeneous earthquake rupture; although neither was fully probabilistic, they should be mentioned here as first steps towards quantifying this uncertainty for inundation hazard. Both also discuss the effect of coseismic displacement on onshore hazard and how this can vary locally, as discussed on P3L2. Here (P3L2) the discussion could be expanded to provide greater justification to your methodological approach to near field hazard.*

These references provide as said a broader context to the paper and we already listed above those we will include in the revised manuscript.

We will improve the introduction accordingly, following all your suggestions, for the different categories, such as: the uncertainty quantification, the computational approaches, challenges for PTHA inundation, rupture complexity and near-field, coseismic displacement and onshore hazard.

> *P1L20: This isn't true. In practice many inundation assessments also use 'representative scenarios' for a range of return periods, not just 'worst credible'.*

We will clarify the statement, adding the mention to "representative scenarios" and some appropriate references - listed above as well, mostly using some scenarios for different representative recurrence times sometimes combined with worst case ones, that is: Gailler et al. 2015; Harbitz et al. 2012; Løvholt et al. 2006; Brizuela et al., 2014.

> ➤ *P1L22: One or a limited range of inundation scenarios get used for much more than 'a first screening' by emergency managers. These scenarios regularly get used to develop emergency management plans, evacuation plans, undertake impact assessments and so on. In my opinion this paragraph severely underplays the utility of scenario hazard assessments. The main problem is that we can't translate the offshore probability to an onshore probability. I expect that even with probabilistic inundation hazard maps, single event scenarios will still be used for a range of emergency management scenario planning purposes – we'll just be in a position to actually say what the probability of the event in terms of inundation hazard is.*

Here we need to disagree a bit; or better, this is not what we meant, since we also wrote: "to realize very detailed assessments of specific scenarios." This goes beyond the "first screening", in our intention. We will clarify this in the revised manuscript, also referring to disaggregation of PTHA for selecting physically meaningful individual scenarios. Instead, we are sorry but we are not sure we understand the statement "we can't translate the offshore probability to an onshore probability.", since in this paper - as well as in other papers from different authors - fully probabilistic inundation maps are presented.

> ➤ *P2L20: Need to clarify that this is talking about onshore PTHA – offshore PTHA are in general computationally affordable (though not cheap!) these days.*

Agree, we will modify the text accordingly.

> ➤ *P2L30: 'while solving all the emerging technical and scientific issues'. This seems a fairly bold claim! Perhaps rephrase.*

We apologise for the misunderstanding: it was intended to emphasize that the work also concerned the implementation of the procedure, which was not trivial. We will rephrase according to your comment.

*2. Method outline This section is clear and well-written*

*3. Implementation of an improved filtering methodology*

> ➤ *P5L4-5: How confident are you in the assumption that similar wave heights lead to similar onshore hazard? What about other wave properties such as period, which may be significant in determining onshore behaviour. E.g. Satake et al 2013 showed how inundation from the Tohoku tsunami was variably controlled by long-period components on flat coastal plains and shorter-period peaks in steep coastal areas. While set within a tsunami warning context rather than hazard assessment context, Gusman et al 2014 used two cycles of a tsunami waveform in identifying 'similar' tsunami. I think some of the issues are resolved for near field tsunami in your coseismic deformation filtering approach presented following, but it could still be good to comment on this issue here.*

We generally agree and we have responded to the related general comment 3. above. We nevertheless give some specific answers here, partly repeating our previous answer.

The general assumption that, for a given source, offshore tsunami amplitude profiles are representative of the coastal inundation behind was applied and tested in the previous work by Lorito et. al (2015). On the other hand, we agree that caution must be used as well, since the previous paper did not deepen into any possible specific case.

Indeed we faced for example the problem when treating near field scenarios, as you observed.

We also agree that there might be other issues. As said the wave period is an important property controlling the tsunami impact: in fact, we someway accounted for that by considering a control profile along the target coast, advancing a kind of ergodic hypothesis.

Future developments of our method could take into account a Gusman-like approach, considering the tsunami time history at each point of the control profile, instead of just the maximum wave height. We will nevertheless add a few comments about this in the manuscript adopting the suggested line of reasoning. We thank you for pointing this out.

> *P6L30-35: It is not entirely clear how the distance is measured across the grid of coseismic deformation points, and how the spatial component is handled – perhaps also write the relevant equation to ensure clarity.*

The comparison between the coseismic deformation fields is carried out point-to-point. The squared Euclidean distance is the metric used for the cluster analysis and only the vertical components are taken into account. We will try and rephrase for the sake of clarity.

*4. The Milazzo oil refinery*
> *P8L28: The abbreviation Mmax is very commonly used to mean the maximum magnitude for a given earthquake source in seismic and tsunami hazard assessment. I would suggest changing this to something else to avoid confusion.*

We will change this MFmax (maximum momentum flux). This will appear as well in the new Figures (those after correcting the results for the bug) in the Supplementary Materials.

> *P9L11: This should be 'overestimates the probability for a given Hmax relative to STEP (3b).*

Ok, we will modify the sentence as suggested.

> *P9L23: Should these be >=, not =, if you're talking about probabilities of exceedance.*

Probability maps are obtained by vertically "cutting" the hazard curves for each point of the grid, i.e. representing on a map the exceedance probability values for a fixed $H_{max}$. In this

sense, the "=" sign is correct. We will add the "(exceedance)" in parentheses before probability for clarity.

> *P9L26-30: Use of phrase 'positive' and 'negative differences' is confusing and makes the meaning of the paragraph somewhat ambiguous. Better to rephrase stating more explicitly which model gives relatively higher/lower hazard etc.*

We agree that the sentences are quite unclear; we will rephrase them referring more explicitly to higher/lower hazard.

> *Also, the difference between results far inland, near the coast and offshore in Figure 4a need to be explained. Why the shift from negative to positive differences at some distance inland from the coast?*

We first need to point out that Figure 4 has changed based on the new results.

We assume that you are referring to the Figure with the differences between the probability maps for steps 3a and 3b and for the intensity threshold of 2 meters (the top right one). We apologise for the confusion and we will add labels where missing to all Figures.

However, note that now this Figure is quite different from before, as it contains more positive values. This is consistent with the new Figure 3c, where the differences are already positive for this threshold and even larger for the 3 metres threshold.

As far as the negative inland values are concerned, note that they occur for very low probability values. So, maybe they shouldn't be overinterpreted. We will however comment all the new Figures based on the new results in the revised manuscript.

> *P10L5: Can anything additional be said about possible biases in the sampling process? Why is it likely that the sampling produced a non-representative selection of the important scenarios? How does this overall affect the strength of you conclusions in comparing the two methods (i.e. could the differences be random rather than systematic).*

What we meant here is that without the correction for near field, namely without a separate treatment for remote and local sources, the filtering procedure provides a non-representative selection of the important scenarios.

This is due to the basic assumption that offshore tsunami amplitudes can be considered representative of the onshore tsunami impact, which surely introduces a bias when the scenarios which deform the coastline are not separated by those that doesn't do it (step (3a)).

For example, admit that 2 different scenarios will both produce 1 meter wave offshore. However, one scenario uplifts the coast of 1 meter, the other one creates 1 meter

subsidence. The two inundations will be dramatically different, but the two scenarios would be nevertheless grouped by (3a) under the same cluster.

Therefore, the procedure proposed as step (3b) is in principle the correct one to evaluate site-specific tsunami hazard, when local effects of coseismic deformation can not be neglected.

5. Conclusions

> *P10L10: The statement around the definition of the source scenarios seems a bit strong. I'd suggest removing the word 'fully' as I doubt this has really been done. Aleatory uncertainty applies to both the rate model and the source location, geometry, maximum magnitude etc. I'd suggest putting 'and their mean annual rates' prior to 'exploring source uncertainty'.*

We agree with the comment: the word "fully" here is misleading. Although in principle the proposed methodology allow us for a full exploration of the aleatory uncertainty, some practical limitations are always present in real life. We will correct the text accordingly.

> *P10L19: Suggest 'from offshore wave amplitudes alone'. Also, what about other parameters such as period for non near-field tsunami? This links back to my comments on Section 3.*

Ok, got it, we'll rephrase by saying only that it is unlikely that the assumption holds if there is co-seismic coastal displacement.

*Figures:*
> *Figure 1: Step 2 should read 'tsunami propagation to offshore points'*
> *Figures 3-5 need labels for parts a), b) etc.*

OK, we added labels where missing in the new figures.

**Technical corrections**
> Throughout: Why use STEP instead of Step?
> P1L11: demonstrate not demonstrated
> P2L25: Rephrase to 'This allows identification of a subset of ...'
> P2L29: Rephrase to 'Here we merge the two approaches of Lorito et al....'
> P3L21: Change 'resume' to 'summarise'. Also P10L9
> P4L8: Change 'enough representative' to representative enough'
> P6L10: Change 'and a separate modelling' to 'and separate modelling'
> P6L34-35: Change to 'while the stopping criterion is based on the Euclidean distance'
> P7L16: Mediterranean Sea (not sea)
> P8L1: Replace 'Namely' with 'That is'; delete 'even'
> P8L25: Please specify the shear modulus used for the Okada calculations

- ➢ P9L15: Remove 'supposedly'
- ➢ P10L24: Change 'has not to be' to 'is not'.
- ➢ P11L3-4: I think this should read 'As a consequence, the effect of coastal deformation on tsunami hazard can not be deduced...'
- ➢ P11L14: Change to '...the approach developed here allows consideration of a very high number...'

We thank you for these technical corrections, which will be all addressed in the revised manuscript. Concerning the Okada calculations, a poissonian solid is assumed with λ=μ, so that the Okada results are independent of the shear modulus.

---

## Author Comment (AC3) · 25 Oct 2018

Dear Reviewer,

we thank you for your thoughtful comments. We addressed all of them as specified in detail in the point-to-point answers in the supplement pdf file, both in response to the general and to the specific comments of yours.

Here, we make some general remarks, since we made one important change in the revised manuscript. We also ask for a minor change to the title to insert in it the word "tsunami", which was missing in the original title. This letter is repeated in all the three

responses to the three Reviewers.

Changes to the results and to the manuscript.

We first of all need to point out that the change we made was necessary, since we found a bug in one of the numerical codes we had written for this study. This bug was found while performing some tests, some of them conceived for addressing your comments, particularly as far as the robustness and the importance of the correction for the near-field sources compared to the "noise" introduced by the tuning of the various filtering thresholds were concerned.

The bug consisted in a missing sum operator in the computation of the cluster probability (a missing cycle over one variable!). Hence, the probability of the entire cluster was not assigned to the cluster representative.

The new results, computed after the bug was corrected, do not differ in essence, although the resulting probabilities are obviously overall higher. All the new Figures are enclosed.

Conversely, for hazard intensities higher than 1 meter, the results now show even more pronounced differences between the "corrected" and "uncorrected" filtering procedures (new Figure 3c).

Our results now more clearly point out that not considering an appropriate correction for the near field would lead to overestimate the tsunami hazard. This is true in the case of this specific setting, though, since we found a prevalence of clusters causing coastal uplift from the near-field sources (the situation may be the opposite as well or a mix for different source-coast configuration).

These uplifts would tend to diminish tsunami inundation. Hence, the tsunami hazard would be overestimated without taking this into account. We hope this is illustrated by the new Figure 3c.

We now in fact added to panel c of this Figure a new quantity, that is the Mean Uplift

(hereinafter MU) on a random point on the coastline in the inner - highest resolution - grid domain.

The MU provides the mean - over all scenarios contributing to some hazard intensities and all coastal points - co-seismic coastal displacement (with positive sign if uplift) plotted versus different hazard intensities, and it is compared to the relative percentage differences between the corrected and uncorrected results.

In more detail, the MU is obtained:

1. for the mean model - the same considered before as far as epistemic uncertainty is concerned;

2. for each single hazard intensity threshold, as said;

3. by performing a weighted average of the uplifts from each model (represented through the centroid of the cluster), where the weights are the annual probabilities of the individual models (of the individual earthquakes then), set to zero if the earthquake do not deform the coastline (i.e. for far-field sources) or if the tsunami doesn't exceed the given hazard intensity threshold; the weighted average is normalized to the total probability of the near- and far-field sources contributing to the tsunami hazard for that threshold;

4. by further averaging the result along the coastline, hence the MU may be interpreted as the mean on a random point of the coastline, from all the far-field and near-field scenarios, the latter including those causing both subsidence and uplift (note than that the absolute MU value in meters is then rather small being averaged over sources that cause either uplift or subsidence, or no coastal displacement at all).

Note that the intermediate results (before applying item 4.), that is the MU on each coastal point, for different intensity thresholds, both for single cluster representatives (red lines) and for the weighted average (according to item 3., blue line) are plotted in the new Figure S4. While we note that there are both positive and negative displacements (red lines corresponding to uplift and subsidence along the coast respectively), the predominant one is unveiled by the sum over the different clusters plotted (the a blue line).

Moreover, the results in Figure 3c now show very little differences between the "corrected" and "uncorrected" filtering procedures at low hazard intensities, that is those below the Filter H thresholds value of 1 meter.

In summary, for the specific case study, that is for this specific source-target configuration, our findings show that not considering an appropriate correction for near field would lead to overestimate the tsunami hazard for Hmax greater than 1m, and this overestimation is correlated to dominant coastal uplift. At lower intensities differences are small but not meaningful, as the results are biased by Filter H.

We will of course add the necessary new text in the revised manuscript concerning MU and the corresponding analysis.

New title.

We propose the following new title for this study: "From regional to local SPTHA: efficient computation of probabilistic tsunami inundation maps addressing near-field sources" That is, we just inserted the word "tsunami" before inundation. This would make easier to find the article if searching NHESS for tsunami-related papers.

Kind Regards, Manuela Volpe, on the behalf of the co-authors.

Please also note the supplement to this comment:
https://www.nat-hazards-earth-syst-sci-discuss.net/nhess-2018-202/nhess-2018-202-AC3-supplement.pdf
* * *
[Figure]

[Figure]

[Figure]

**Fig. 1.** Figure 3

[Figure]

**Fig. 2.** Figure 4

[Figure]

**Fig. 3.** Figure 5

**Fig. 4.** Figure S2

[Figure]

[Figure]

[Figure]

[Figure]

**Fig. 5.** Figure S3

**Fig. 6.** Figure S4

**Supplement:**

**Response point-by-point to Anonymous Referee #3**

The point-by-point answers are in blue color, below each Reviewer's comment (reported in *Italic*).

*Overview: The authors did an extension of the SPTHA method previously proposed by Lorito et al. (2015) [GJI] and Selva et al. (2016) [GJI]. A new filtering scheme for earthquake scenarios is developed (Filter P) and the method is applied to a particular coast (Milazzo).*
*Overall evaluation: The application of the SPTHA to a new region and adding some innovations to the previously-developed method may justify publication of this work*
*The paper is not as good as the other two papers published before (I mean Lorito et al. 2015; Selva et al. 2016). The current manuscript refers to the previous two papers very frequently and does not seem to stand by its own. However, I am positive about this work and I think it can be published in NHESS after some revisions. I made some suggestions below.*

We thank for your positive evaluation. Our manuscript in fact does not propose a totally new method, but develops an upgrade of the method previously proposed by the cited published papers. This is the reason for frequently referring to them.

*Comments:*
- ➢ *Page 3, "Method": your Section 2 looks a review of the methods previously published by Lorito et al. (2015) [GJI] and Selva et al. (2016) [GJI]. Your own method is outlined in Section 3. This is confusing. In fact, your current section 2 is sort of literature review. I suggest change the title of Section 2 to "A review of SPTHA" and then change title of Section 3 to "Methods: an improved SPTHA".*

We understood the point, although Section 2 is not properly a literature review of SPTHA but just a summary of the original method. Anyway, we will provide more suitable titles. For example, they might be: "A review of the original method" for Section 2 and "Improvements in the filtering procedure" for Section 3.

- ➢ *To show the better performance of the new method over the ones published before (I mean Lorito et al. 2015; Selva et al. 2016), a discussion or a figure is needed.*

The improved method, illustrated in Figure 1 of our manuscript, fits into the general scheme displayed in Figure 1 of Selva et al. 2016, who already foresaw the possibility of performing site specific tsunami hazard, although they did not addressed nor implemented or tested it.

The performances of the "new" method step (3b) are here benchmarked with respect to step (3a), which corresponds to the original method as, although including some improvements, it lacks the most crucial novelty, that is the separate treatment of the near-field scenarios. It is discussed in several places that this was not done by Lorito et al. 2015. We will consider if stressing this again while summarizing the results.

➢ *Try not to refer to two previous papers so much. You may want to show more independence.*

Ok, we will remove the references where possible.

➢ *In Page 9, refer to appropriate figures when discussing the results.*

We agree, and will modify accordingly.

➢ *Why you have capital letters for STEP? Is that necessary? If not, change it to "step" because when you use capital letters, the reader assumes it is an acronym. I guess it is not an acronym for anything.*

We did not consider possible confusion with an acronym. As capital letters are not really necessary here, we will change it.

➢ *Page 5, Line 17: explain more about Filter P.*

Filter P is further explained by lines 17-29. We will try to make it clearer in the revised version.

➢ *Page 6, Line 2: what is intra-cluster? It is unclear. Make sure to explain more about it and clarify how it works.*

Intra-cluster variance means the variance within each cluster; it was used in the original method to define the optimal number of cluster in the cluster analysis procedure, according to the so-called Beale test. This will be better explained in the revised text.

➢ *Page 6, Line 7: delete statements like "as mentioned before..." it is not suitable for academic writing.*
➢ *Page 7, Line 32: delete "as discussed in previous ...." Again not suitable.*

We accept the comments and will revise accordingly

➢ *Page 8: here you use "cluster" and "scenario" interchangeably. Make sure which one you meant. I assume that you meant "Scenario" not "cluster". They are different. Cluster is much bigger than a single scenario. One cluster can include 200 scenarios. In Line 13, you say: "We obtained 634 and 520 clusters for remote and local sources, respectively, that is a total of 1154 scenarios ...". Here the sum of 634 and 520 clusters cannot be 1154 scenarios. Instead, the sum of 634 and 520 clusters cannot be 1154 CLUSTERS.*

There is no doubt that "cluster" and "scenario" are two different things. At the end of the cluster analysis we obtain clusters of scenarios, but for each cluster a representative scenario is selected, to which the probability of occurrence of the entire cluster is assigned, and then the representative scenarios are the ones which are explicitly modeled. In this

sense a certain number of clusters corresponds to the same number of scenarios to be simulated. We will clarify in the text.

> ➤ Your conclusion has many repeats; for example lines 8-13. Make sure delete all repeats.

> ➤ ABSTRACT: try to have more numbers and conclusions, not only generic statements.
> ➤ CONCLUSIONS: shorten it to a paragraph and be specific and do not repeat all stuff again.

We will take into account the above general style suggestions in the revised text, thank you.

> ➤ *Figure 1: The last box repeats. Delete one of them.*

The last box, relative to step (4) is repeated twice in order to highlight that step (3a) and (3b) are completely separate paths. In our opinion, reporting step (4) just once, connected both to step (3a) and (3b) is misleading, as it could suggest a "merging" of the results of simulations from the two paths to evaluate SPTHA.

> ➤ *Figure 2: Explain what are two sets of red dots.*

We agree: this information is missing in the figure caption. We will correct.

---

## Author Response (AR2)

Dear Editor, please find below our responses to the Reviewer's comments as well as the marked-up manuscript version, where all the changes we did are evidenced.

Kind Regards,

Manuela Volpe, on the behalf of the co-authors.

Dear Reviewers, we thank you for your comments. We addressed all of them as specified in detail in the point-to-point answers below.

Kind Regards,
Manuela Volpe, on the behalf of the co-authors.

**Response point-by-point to Anonymous Referee #1 (Report #1)**

*This is now in good shape -- but I think the abstract needs slight edits as detailed below.*

***Moreover, we developed a strategy to identify, on the basis of the tsunami initial conditions, and separately treat near-field scenarios, to avoid biases in the tsunami hazard assessment***
*- This needs to be rephrased, it doesn't make sense. What do you "identify"?*
*- Perhaps you mean "Moreover we developed a strategy to identify tsunami initial conditions, and separately ...."*

We apologise for the misunderstanding, as "identify" is referred to "near-field scenarios". We rephrased in the text.

***Therefore, we proposed two parallel filtering schemes in the far and the near-field, based on the similarity of offshore tsunamis and hazard curves and on the coseismic fields, respectively.***
*- I think this needs rewording -- I'm confused about the "implied grouping" of the "offshore tsunamis", "hazard curves", "co-seismic-fields". Consider rephrasing like:*
*- Therefore, we proposed two parallel filtering schemes in the far and the near-field, based on the similarity of offshore tsunamis and 1. hazard curves, 2. coseismic fields, respectively.*

Here there is a typo, as "on the coseismic fields" should be "of the coseismic fields": for better clarity we rephrased this sentence too.

**Response point-by-point to Anonymous Referee #4 (Report #2)**

*In general I think this paper is a good contribution and worth publishing.*

*There is one thing that confuses me, however, that perhaps need to be better explained.*

*In Figure 3(c), the purple line shows the MU, mean uplift on a random point along the coastline. I assume (based on the discussion at the bottom of page 10) that at each point on the coast this is computed by averaging over all scenarios that inundated at least to level Hmax at this point. So I think that as Hmax goes to zero this should approach a constant value, given by the mean uplift over all points and all scenarios. It is not clear to me why this curve is not shown for Hmax < 1m, which would allow one to better test this understanding.*

The mean uplift on a random point along the coastline (MU) is computed as you described but also averaging over all of the coastal points.

We computed such quantity only for hazard intensities greater than 1m since, being this value the adopted threshold for Filter H both in step (3a) and in the far-field-branch of step (3b), results at lower intensities can be biased, as explicitly declared in the text.

Nevertheless, we now computed MU also for 0.1<Hmax<1, as shown in the figure below, where a zoom within the low intensity range is displayed. As you expected, the curve tends to a plateau, which turns out to be slightly negative, suggesting that uplift is not the predominant effect in that range (indeed, the percentage differences between the two approaches are negative as well in that region). We now added this interval in the Figure 3c as well. On the other hand, we restate that such outcomes can not be considered really robust and in our opinion further speculations should be avoided.

[Figure]

*Also, I would think that the scenarios with subsidence at the coast would give the most flooding, not the ones that have uplift. So for larger values of Hmax I would have expected that averaging MU only over the scenarios that exceeded Hmax would tend to give values of MU that are negative and decreasing as Hmax increases. The purple curve does decrease for Hmax > 3m, but it is not clear why it is increasing before this and positive everywhere except at the the largest value Hmax = 6m. The comment about this on page 11, line 8 of the paper says "Anyway, the obtained positive values indicate that the uplift of the coast is prevailing, consistently with the positive percentage differences retrieved between the two approaches for Hmax > 1m." It is not clear to me why this is consistent or expected from the positive differences between the two approaches. Some additional explanation would be useful.*

A positive difference between the two approaches means that, without the correction for near-field scenarios, the tsunami hazard would be overestimated, as the "real" hazard is decreased by coastal uplift, which is predominant in the investigated range (Hmax>1). From Figure S4, it can be seen that near-field scenarios contributing to higher intensities are the ones producing small coseismic displacements. This is because scenarios causing important displacements do not exceed high values of Hmax (i.e. do not contribute to those points of the hazard curves) just because of the coastal uplift. In other words, most flooding can occur also in presence of small uplift, not only subsidence. We added such consideration in the text.

*Another comment concerning Figure 3(a,b) is that the individual hazard curves are impossible to see, as these figures are mostly a mass of gray. In particular there is no way to compare things visually between 3(a) and (b) and so these figures are not very useful. I wonder if there is a way to clarify this, perhaps by only showing the hazard curves at a much smaller number of representative points rather than plotting them for all points?*

You are right; however the first two panels just show how the bulk of hazard curves appear, although a curve by curve comparison is not possible from them. Additional information is in the other two panels of the same figures, where the point-by-point percentage differences are for example plotted. Nevertheless, to provide more information as you suggested, we now added a figure in the supplementary material (also attached below for your convenience), where we compare a sample of curves (one every thousandth) at inland points. The dashed lines correspond to step(3a) while the plain lines to step(3b). The behaviour of each curve clearly depends on the grid point and on the scenarios which contribute. Nevertheless, the sampled curves confirm the overestimation as the dashed lines mostly lie above the continuous ones, at least for the higher intensities.

[Figure]

**From regional to local SPTHA: efficient computation of probabilistic tsunami inundation maps addressing near-field sources**

Manuela Volpe, Stefano Lorito, Jacopo Selva, Roberto Tonini, Fabrizio Romano, and Beatriz Brizuela

Istituto Nazionale di Geofisica e Vulcanologia, Italy

**Correspondence:** M. Volpe (manuela.volpe@ingv.it)

**Abstract.** Site-specific Seismic Probabilistic Tsunami Hazard Analysis (SPTHA) is a computationally demanding task, as it requires in principle a huge number of high-resolution numerical simulations for producing probabilistic inundation maps. We implemented an efficient and robust methodology using a filtering procedure to reduce the number of numerical simulations needed, while still allowing full treatment of aleatory and epistemic uncertainty. Moreover, to avoid biases in the tsunami hazard assessment, we developed a strategy to identify and separately treat tsunamis generated by near-field earthquakes we developed a strategy to identify, on the basis of the tsunami initial conditions, and separately treat near-field scenarios, 
[revised manuscript text omitted]